# PIE: SIMULATING DISEASE PROGRESSION VIA PROGRESSIVE IMAGE EDITING

## ABSTRACT

Disease progression trajectories can greatly affect the quality and efficacy of clinical diagnosis, prognosis, and treatment. However, one major challenge is the lack of longitudinal medical imaging monitoring of individual patients over time. To address this issue, we propose Progressive Image Editing (PIE) method that enables controlled manipulation of disease-related image features, facilitating precise and realistic disease progression simulation in imaging space. Specifically, we leverage recent advancements in text-to-image generative models to simulate disease progression accurately and personalize it for each patient. We also theoretically analyze the iterative refining process in our framework as a gradient descent with an exponentially decayed learning rate. To validate our framework, we conduct experiments in three medical imaging domains. Our results demonstrate the superiority of PIE over existing methods such as Stable Diffusion Video and Style-Based Manifold Extrapolation based on CLIP score (Realism) and Disease Classification Confidence (Alignment). Our user study collected feedback from 35 veteran physicians to assess the generated progressions. Remarkably, 76.2% of the feedback agrees with the fidelity of the generated progressions. PIE can allow healthcare providers to model disease imaging trajectories over time, predict future treatment responses, fill in missing imaging data in clinical records, and improve medical education. [1]

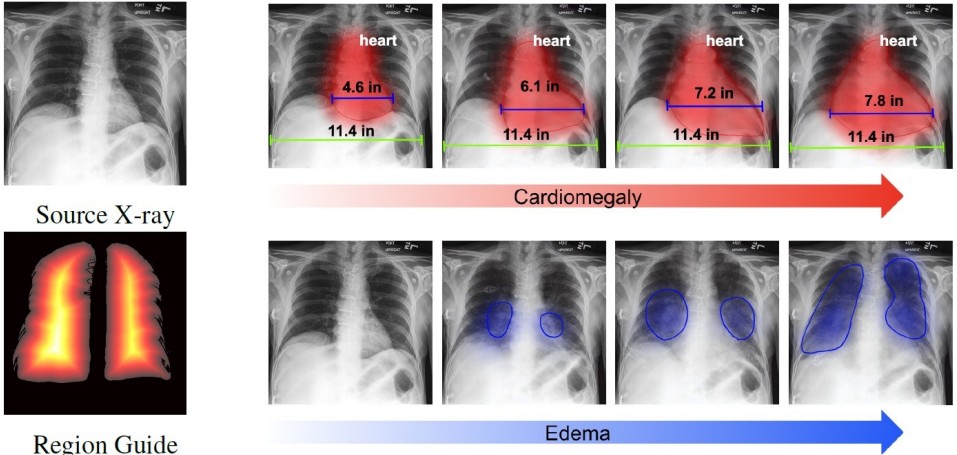

Figure 1: Illustrative examples of disease progression simulation using PIE. The top progression sequence depicts a patient's heart increasing in size (red), indicating Cardiomegaly. The bottom sequence demonstrates the expanding mass areas (blue) in a patient's lung, indicating Edema.

---

[1]The reproducibility and ethics statement are included in the supplemental material D and G. Anonymous code for replicating our results can be found at anonymous.4open.science/r/PIE-3332.

## 1 INTRODUCTION

Disease progression refers to how an illness develops over time in an individual. By studying the progression of diseases, healthcare professionals can create effective treatment strategies and interventions. It allows them to predict the disease's course, identify possible complications, and adjust treatment plans accordingly. Furthermore, monitoring disease progression allows healthcare providers to assess the efficacy of treatments, measure the impact of interventions, and make informed decisions about patient care. A comprehensive understanding of disease progression is essential for improving patient outcomes, advancing medical knowledge, and finding innovative approaches to prevent and treat diseases.

However, disease progression modeling in the imaging space poses a formidable challenge primarily due to the lack of continuous monitoring of individual patients over time and the high cost to collect such longitudinal data (Sukkar et al., 2012; Wang et al., 2014; Liu et al., 2015; Cook & Bies, 2016; Severson et al., 2020). The intricate and multifaceted dynamics of disease progression, combined with the lack of comprehensive and continuous image data of individual patients, result in the absence of established methodologies (Hinrichs et al., 2011; Ray, 2011; Lee et al., 2019). Moreover, disease progression exhibits significant variability and heterogeneity across patients and disease sub-types, rendering a uniform approach impracticable.

Past disease progression simulation research has limitations in terms of its ability to incorporate clinical textual information, generate individualized predictions based on individualized conditions, and utilize non-longitudinal data. This highlights the need for more advanced and flexible simulation frameworks to accurately capture the complex and dynamic nature of disease progression in imaging data. To incorporate the generation model into a conditioned simulation of disease progression, we propose a progressive framework PIE, for disease progression simulation that combines text and image modalities. Specifically, we aim to progressively add and subtract disease-related features, controlled by a text encoder, to conditionally progress the disease without significantly altering the original base image features (see Figure 1). Our framework is built based on the invertibility of denoising diffusion probabilistic models (Ho et al., 2020; Song et al., 2020a). Our theoretical analysis shows PIE can be viewed as a gradient descent toward the objective maximum log-likelihood of given text conditioning. The learning rate in this iterative process is decaying exponentially with each iteration forward, which means that the algorithm is effectively exploring the solution space while maintaining a balance between convergence speed and stability. This theoretical analysis guarantees that our framework is moving the instance toward the targeted manifold and ensures modification is bounded.

We evaluate PIE on three distinct medical imaging datasets with non-longitudinal disease progression data, including Chexpert (Irvin et al., 2019), Diabetic Retinopathy Detection (CHF, 2015) and ISIC 2018 (Codella et al., 2019). We demonstrate that our framework leads to more accurate and individualized disease progression predictions on these datasets, which can improve clinical diagnosis, treatment planning, and enhance patient records by filling in missing imaging data and potentially helping medical education. We also conducted a user study with physicians to evaluate the effectiveness of PIE for disease progression simulation. The study presented physicians with a set of simulated disease images and progressions, and then asked them to assess the accuracy and quality of each generated image and progression.

- We propose a temporal medical imaging simulation framework PIE, which allows for more precise and controllable manipulation of disease-related image features and leads to more accurate and individualized longitudinal disease progression simulation.

- We provide theoretical evidence that our iterative refinement process is equivalent to gradient descent with an exponentially decaying learning rate, which helps to establish a deeper understanding of the underlying mechanism and provides a basis for further improvement.

- We demonstrate the superior performance of PIE over baselines in disease progression prediction with three medical domains. The results show that PIE produces more accurate and high-quality disease progression prediction.

- We also conducted a user study with physicians to evaluate the effectiveness of our proposed framework for disease progression simulation. The physicians agree that simulated disease

progressions generated by PIE closely matched physicians' expectations $76.2\%$ of the time, indicating high accuracy and quality.

## 2    RELATED WORKS

**Disease Progression Simulation** Longitudinal disease progression data derived from individual electronic health records offer an exciting avenue to investigate the nuanced differences in the progression of diseases over time (Schulam & Arora, 2016; Stankeviciute et al., 2021; Chen et al., 2022; Mikhael et al., 2023; Koval et al., 2021). Most of the previous works are based on HMM (Wang et al., 2014; Liu et al., 2015; Alaa et al., 2017) and deep probabilistic models (Alaa & van der Schaar, 2019). Some recent works start to resolve disease progression simulation by using deep generation models. (Ravi et al., 2022) utilized GAN-based model and linear regressor with individual's sequential monitoring data for Alzheimer's disease progression simulation in MRI imaging space. However, all these methods have to use full sequential images and fail to address personalized healthcare in the imaging space. The lack of such time-series data, in reality, poses a significant challenge for disease progression simulation (Xue et al., 2020; Chen, 2022; Berrevoets et al., 2023).

**Generative Models** Generative models like Variational Autoencoders (VAEs) (Kingma & Welling, 2013) and Generative Adversarial Networks (GANs) (Goodfellow et al., 2020) have been widely employed in medical imaging applications (Nie et al., 2017; Isola et al., 2017; Cao et al., 2020). Recent GAN models  (Kang et al., 2023; Patashnik et al., 2021) have harnessed the power of CLIP (Radford et al., 2021) embedding to guide image editing based on contextual prompts. However, GAN-based models are unstable and difficult to optimize in general. Denoising Diffusion Models (Sohl-Dickstein et al., 2015; Ho et al., 2020; Song et al., 2020a; Rombach et al., 2022; Karras et al., 2022) have become increasingly popular in recent years due to their ability to create photo-realistic images from textual descriptions. One major advantage of these models is their ability to learn from large-scale datasets.  Among the various text-to-image models, Stable Diffusion  (Rombach et al., 2022) has received considerable attention because of its impressive performance in generating high-quality images and its relatively low cost to fine-tune. Its denoising process works similarly to the diffusion models but in a latent space, and this process results in a final image that is highly consistent with the input text, making it an excellent tool for text-guided image editing. Diffusion models can also be effortlessly incorporated into an image-to-image editing pipeline (Brooks et al., 2022; Parmar et al., 2023; Orgad et al., 2023), thus providing users the ability to edit scenarios across multiple modalities and assess potential imaging progressive editing paths. However, existing image-to-image methods can only be used for single-step editing, which makes it difficult to simulate personalized time-series progression data in the medical domain.

## 3    PROBLEM STATEMENT

In the traditional disease progression simulation setting, assume having sequential time-series image-text data pairs $\{(\boldsymbol{x_0}, y_0), (\boldsymbol{x_1}, y_1), ..., (\boldsymbol{x_T}, y_T)\}$ from each patient. The clinical image-text data pair $(\boldsymbol{x}, y) \in \mathcal{X} \times \mathcal{Y}$ is sampled from a non-independent identically distribution, where $\mathcal{Y} = \mathbb{R}^n$ denote the medical report space and $\mathcal{X} = \mathbb{R}^m$ denote the medical imaging space. All the prior works either rely heavily on probability modeling: $f_\theta(y_{0:t-1}) \to y_t$ (Liu et al., 2015; Alaa & van der Schaar, 2019), or rely on using longitude data to train regression models for imaging simulation: $f_\theta(x_{0:t-1}, y_{t-1}) \to y_t$ (Han et al., 2022; Ravi et al., 2022). However, it is hard to obtain sequential longitude data as most patients may not go to the same hospital for follow-up treatment. And the hospitals also lack medical imaging and clinical reports in the early stages of the disease.

In this paper, we redefine disease progression simulation using a data-driven generative model without the need for sequential time-series data or clinical prior knowledge. Anyone with access to discrete imaging and medical report data could individually train the model to predict disease progressions without profound medical prior, significantly reducing the amount of work required for feature engineering and data collection.

**Definition 1 (Simulate disease progression with non-sequential data)** *Assume $h_\phi$ is a generative model learned from the data space: $\Omega = \{(\boldsymbol{x}, y) \in \chi \times \Gamma\}$, assuming it is independent identically distributed and each $(\boldsymbol{x}, y)$ is from different individuals. In training phase, $h_\phi$ models the mapping:*

$\Gamma \to \chi$. *In the inference phase, given an initial test data sample $(x_t, y_t)$ at progression stage $t$, $h_\phi$ converts input imaging $x_t$ and input clinical context $y_T$ to $x_T$, where $y_T$ is the language model inferred final step clinical report from $y_0$ and $x_t, x_{t+1}, ..., x_T$ is the simulated sequential imaging progression.*

In the following sections, we picked DDIM as a base step of our proposed method, because of its reversible theoretical properties that allow smooth transitions and convergence based on Definition 1. The proof is shown in the supplementary section.

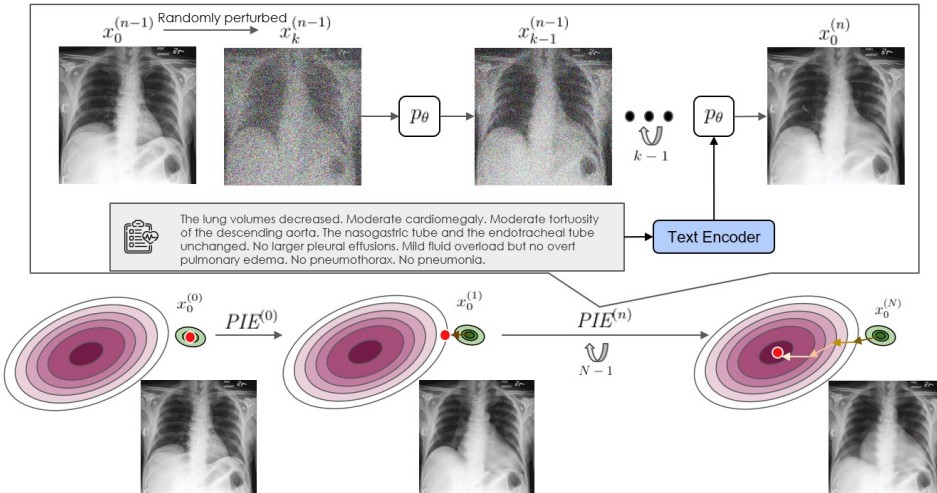

Figure 2: Overview of the PIE inference pipeline. PIE is illustrated using an example of disease progression editing X-ray from a healthy state to cardiomegaly. For any given step $n$ in PIE, we first utilize DDIM inversion to procure an inverted noise map. Subsequently, we denoise it using clinical reports imbued with progressive cardiomegaly information. The output of DDIM denoising serves as the input for step $n + 1$, thus ensuring a gradual and controllable disease progression simulation. After simulating $N$ steps, the image is converged to the final state.

## 4 PROGRESSIVE IMAGE EDITING (PIE)

Progressive image editing (PIE) is a novel framework proposed to refine and enhance images in an iterative and discrete manner, allowing the use of additional prompts for small and precise adjustments to simulate semantic modification while keeping realism. Unlike traditional image editing techniques, PIE involves a multi-stage process where each step builds upon the previous one, with the aim of achieving a final result that is more refined and smooth than if all changes were made at once. The approach also enables precise control over specific semantic features of the image to be adjusted without significant impacts on other regions. The main purpose of PIE is to simulate disease progression from multi-modal input data.

**Procedure.** The inputs to PIE are a discrete medical image $x_0^{(0)}$ depicting any start or middle stage of a disease and a corresponding clinical report latent $y$ as the text conditioning (Rombach et al., 2022). $y$ is generated from a pretrained text encoder from CLIP (Radford et al., 2021) [clip-vit-large-patch14], where the raw text input could either be a real report or synthetic report, providing the potential hint of the patient's disease progression. The output generated is a sequence of images, $\{x_0^{(0)}, x_0^{(1)}, ..., x_0^{(N)}\}$, illustrating the progression of the disease as per the input report. The iterative PIE procedure is defined as follows:

**Proposition 1** *Let $x_0^{(N)} \sim \chi$, where $\chi$ is distribution of photo-realistic images , $y$ be the text conditioning, running $PIE^{(n)}(\cdot, \cdot)$ recursively is denoted as following, where $N \geq n \geq 1$,*

$$x_0^{(n)} = PIE^{(n)}(x_0^{(n-1)}, y) \tag{1}$$

*Then, the resulting output $x_0^{(N)}$ maximizes the posterior probability $p(x_0^{(N)} | x_0^{(0)}, y)$.*

With each round of editing as shown in Figure 2, the image gets closer to the objective by moving in the direction of $-\nabla \log p(x|y)$. Due to the properties of DDIM, the step size would gradually decrease with a constant factor. Additional and more detailed proofs will be available in Supplementary B.

**Proposition 2** *Assuming $\|x_0^{(0)}\| \leq C_1$ and $\|\epsilon_\theta(x, y)\| \leq C_2$, $(x, y) \in (\chi, \Gamma)$, for any $\delta > 0$, if*

$$n > \frac{2}{\log(\alpha_0)} \cdot (log(\delta) - C) \tag{2}$$

*then,*

$$\|x_0^{(n+1)} - x_0^{(n)}\| < \delta \tag{3}$$

*where, $\lambda = \frac{\sqrt{\alpha_0 - \alpha_0\alpha_1} - \sqrt{\alpha_1 - \alpha_0\alpha_1}}{\sqrt{\alpha_1}}$, $\chi$ is the image distribution, $\Gamma$ is the text condition distribution , and $C = \log((\frac{1}{\sqrt{\alpha_0}} - 1) \cdot C_1 + \lambda \cdot C_2)$*

**Proposition 3** *For all $N > 1$, $\|x_0^{(N)} - x_0^{(0)}\| \leq [(\frac{1}{\sqrt{\alpha_0}} - 1) \cdot C_1 + \lambda \cdot C_2]$*

In addition, Proposition 2 and 3 show as $n$ grows bigger, the changes between steps would grow smaller. Eventually, the difference between steps will get arbitrarily small. Hence, the convergence of $PIE$ is guaranteed and modifications to any inputs are bounded by a constant.

---

**Algorithm 1:** Progressive Image Editing $n$-th step ($PIE^{(n)}$)

---

**Input:** Original input image $x_0^{(0)}$ at the start point, input image $x_0^{(n-1)}$ at stage $n$, number of diffusion steps $T$, text conditional vector $y$, noise strength $\gamma$, stable diffusion parameterized denoiser $\epsilon_\theta$, a ROI mask $M_{ROI}$, $M_{ROI}^{i,j} \in [0, 1]$

**Output:** Modified image $x'$ as $x_0^n$

1   $x' \leftarrow x_0^{(n-1)}$
2   $k \leftarrow \gamma \cdot T$
3   $\epsilon \sim \mathcal{N}(0, \mathcal{I})$
4   $x' \leftarrow \sqrt{\alpha_k} \cdot x' + \sqrt{1 - \alpha_k} \cdot \epsilon$
5   **for** $t = k$ **to** 1 **do**
6     $x' \leftarrow \sqrt{\alpha_{t-1}}(\frac{x' - \sqrt{1-\alpha_t}\epsilon_\theta^{(t)}(x', y)}{\sqrt{\alpha_t}}) + \sqrt{1 - \alpha_{t-1}} \cdot \epsilon_\theta^{(t)}(x', y)$
7   **end**
8   $x' \leftarrow (\beta_1 \cdot (x' - x_0^{(0)}) + x_0^{(0)}) \cdot (1 - M_{ROI}) + (\beta_2 \cdot (x' - x_0^{(0)}) + x_0^{(0)}) \cdot M_{ROI}$
9   **return** $x'$ **as** $x_0^{(n)}$

---

## 5   Experiments and Results

In this section, we present experiments on various disease progression tasks. Experiments results demonstrate that PIE can simulate the disease-changing trajectory that is influenced by different medical conditions. Notably, PIE also preserves unrelated visual features from the original medical imaging report, even as it progressively edits the disease representation. Figure 5 showcases a set of disease progression simulation examples across three distinct types of medical imaging. Details for Stable Diffusion fine-tuning, pretraining model for confidence metrics settings are available in Supplementary D.

### 5.1   Experimental Setups

**Implementation Details.** We present the details of single-step PIE in Algorithm 1. For $PIE^{(n)}$, we define $\alpha_k$ according to the DDIM case. Line 8 in Algorithm 1 ensures progressive and limited modifications between the original input image $x_0^{(0)}$, the single-step edited output $x'$, and the region guide selector $M_{ROI}$ through the utilization of interpolation average parameters $\beta_1$ and $\beta_2$. These

Table 1: Comparisons with multi-step editing simulations. The backbone of PIE and baseline approaches are Stable Diffusion with the same pre-trained weight.

| Method | Chest X-ray | | Retinopathy | | Skin Lesion Image | |
|---|---|---|---|---|---|---|
| | Conf ($\uparrow$) | CLIP-I ($\uparrow$) | Conf ($\uparrow$) | CLIP-I ($\uparrow$) | Conf ($\uparrow$) | CLIP-I ($\uparrow$) |
| Stable Diffusion Video | 0.389 | 0.923 | 0.121 | 0.892 | 0.201 | 0.886 |
| Extrapolation | 0.0543 | **0.972** | 0.0742 | 0.991 | 0.226 | 0.951 |
| PIE | **0.690** | 0.968 | **0.807** | **0.992** | **0.453** | **0.958** |

parameters dictate the modification ratio between the ROI mask-guided space and the original input space. As $\beta_1$ increases, the multi-step editing process becomes smoother, though it may sacrifice some degree of realism.

**Datasets for Disease Progression.** We validate the disease progression analysis through end-to-end medical domain-specific image inference. Specifically, we evaluate the pretrained domain-specific stable diffusion model on three different types of disease datasets in classification tasks: CheXpert for chest X-ray classification (Irvin et al., 2019), ISIC 2018 / HAM10000 (Codella et al., 2019; Tschandl et al., 2018) for skin cancer prediction, and Kaggle Diabetic Retinopathy Detection Challenge (CHF, 2015). Each of these datasets presents unique challenges and differ in scale, making them suitable for testing the robustness and versatility of PIE. We also collected over 30 healthy data among the test set from these datasets. These data were used for disease progression simulation. Three groups of progression visualization results can be found in Figure 5.

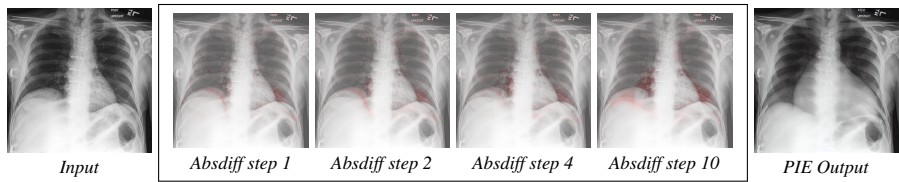

*Input*     *Absdiff step 1*     *Absdiff step 2*     *Absdiff step 4*     *Absdiff step 10*     *PIE Output*

Figure 3: Cardiomegarly disease progression absolute difference heatmap simulated by PIE. The highlighted red portion illustrates the progression of the pathology at each step.

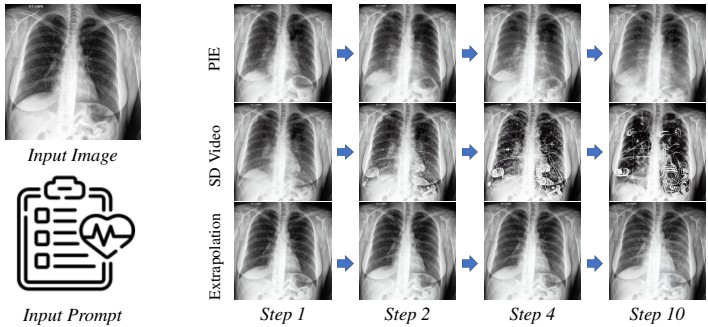

Figure 4: Using PIE, SD Video, Extrapolation to simulate Edema progression with clinical reports as input prompt.

**Evaluation Metrics.** The assessment of generated disease progression images relies on two crucial aspects: alignment to edited disease feature and subject fidelity. To measure these characteristics, we utilize two primary metrics: the CLIP-I score and the classification confidence score. The CLIP-I score (range from [-1, 1] in theory ) represents the average pairwise cosine similarity between the CLIP embeddings of generated and real images (Radford et al., 2021; Ruiz et al., 2022). The classification confidence score is determined using supervised train deep networks for binary classification between negative (healthy) and positive (disease) samples. It is denoted as **Conf** $= Sigmoid(f_\theta(x))$ and represent whether the simulation results are aligned to target disease. In our experiments, we train the DeepAUC maximization method (Yuan et al., 2021) (SOTA of Chexpert and ISIC 2018 task 3) using DenseNet121 (Huang et al., 2017) as the backbone to compute the classification confidence score.

**Baselines** To our knowledge, there are no existing image editing models specifically designed for simulating disease progression without sequential training data. To underscore the unique strengths of PIE, we compare it against two of the most promising state-of-the-art baseline methods. One of them is Stable Diffusion Video (SD Video) (Raw, 2022) for short video generation. SD Video is the code implementation based on recent latent-based video generation methods (Blattmann et al., 2023; Wu et al., 2022). Another one is the Style-Based Manifold Extrapolation (Extrapolation) (Han et al., 2022) for generating progressive medical imaging, as it don't need diagnosis labelled data (Ravi et al., 2019; Han et al., 2022), which is similar to PIE's definition setting but need progression inference prior. During the comparison, all baseline methods are using the same Stable Diffusion finetuned weights and also applied $M_{ROI}$ for region guided.

## 5.2 PROGRESSION SIMULATION COMPARISON

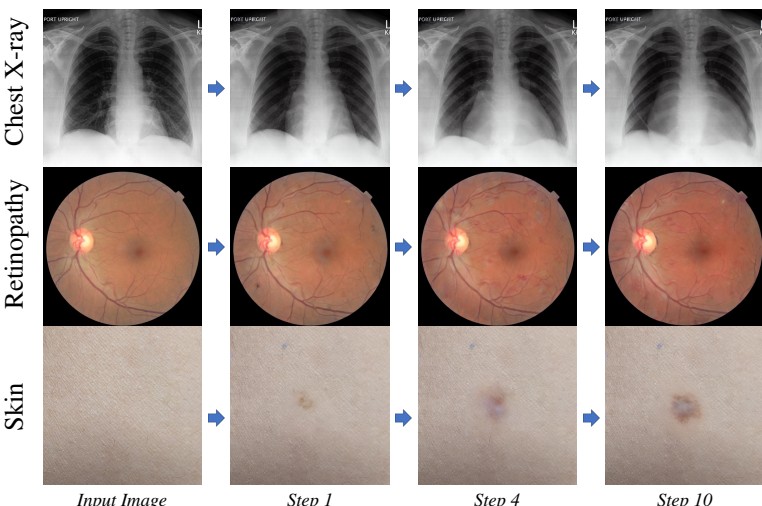

Figure 5: Disease Progression Simulation of PIE. The top progression is for Cardiomegarly. The middle progression is for Diabetic Retinopathy. The bottom progression is for Melanocytic Nevus.

In order to demonstrate the superior performance of PIE in disease progression simulation over other single-step editing methods, we perform experiments on three datasets previously mentioned. For each disease in these datasets, we used 10 healthy samples in the test set as simulation start point and run PIE, SD Video, Extrapolation with 5 random seeds. We obtain at least 50 disease imaging trajectories for each patient. Table 1 showcases that PIE consistently surpasses both SD Video and Extrapolation in terms of disease confidence scores while maintaining high CLIP-I scores. For Chexpert dataset, the 0.690 final confidence score is the average score among 5 classes. For Diabetic Retinopathy and ISIC 2018 datasets, we compare PIE with SD Video, Extrapolation for editing image to the most common seen class since these datasets are highly imbalanced. Figure 6 illustrates the evolution of disease confidence scores during the progression simulation in each step. We observe that PIE is able to produce more faithful and realistic progressive editing compared to the other two baselines. Interestingly, while the CLIP-I score of Extrapolation is comparable to that of PIE, it fails to effectively edit the key disease features of the input images as its confidence scores are low throughout and at the end of the progression. We also visualize the absolute differences between initial stage and each progression stage of Cardiomegaly in Figure. 3.

Figure 4 showcases a group of progression simulation results for Edema in chest X-rays with CheXpert clinical report prompt. It is evident from our observations that while SD Video can significantly alter the input image in the initial step, it fails to identify the proper direction of progression in the manifold after a few steps and would easily create uncontrollable noise. Conversely, Extrapolation only brightens the Chest X-ray without making substantial modifications. PIE, on the other hand, not only convincingly simulates the disease trajectory but also manages to preserve unrelated visual features from the original medical imaging. Further visual comparisons among different datasets are presented in Supplementary E.

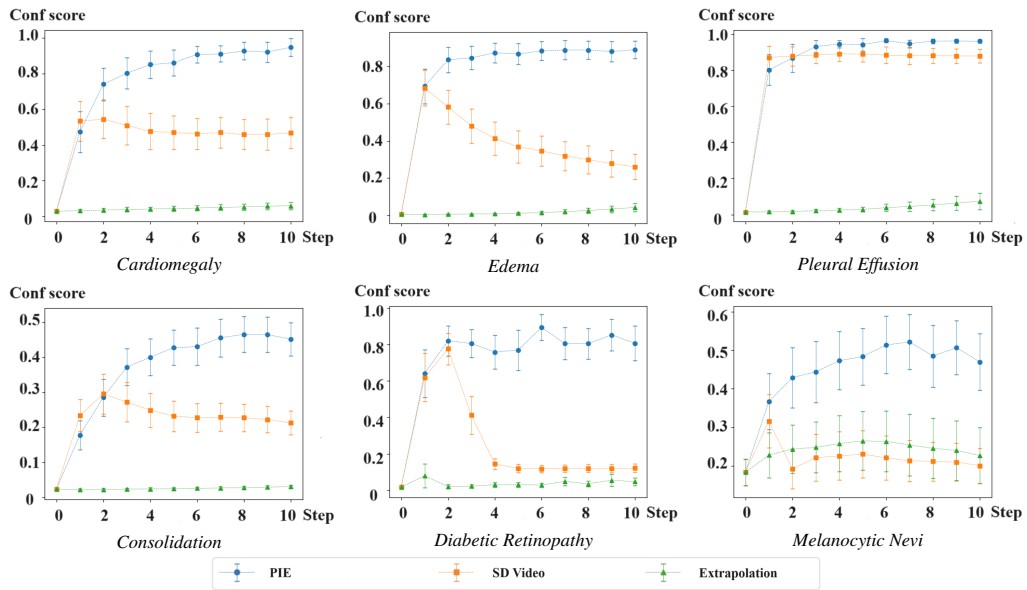

Figure 6: PIE excels in comparison to all the baseline methods across six different disease progression simulations. The inputs utilized are genuine healthy images from the test sets. For each image, we apply five random seeds to simulate disease progression over ten steps. The confidence score, a value that ranges from 0 to 1, signifies the classification confidence for a specific disease.

## 5.3 ABLATION STUDY

**Medical heuristic guidance.** During the PIE simulation, the region guide masks play a big role as prior information. Unlike other randomly inpainting tasks (Lugmayr et al., 2022), ROI mask for medical imaging can be extracted from real or synthetic clinical reports Boag et al. (2020); Lovelace & Mortazavi (2020) using domain-specific Segment Anything models (Kirillov et al., 2023; Ma & Wang, 2023). It helps keep unrelated regions consistent through the progressive changes using PIE or baseline models. In order to generate sequential disease imaging data, PIE uses noise strength $\gamma$ to control the influence from the patient's clinically reported and expected treatment regimen at time $n$. $N$ is used to control the duration of the disease occurrence or treatment regimen. PIE allows the user to make such controls over the iterative process, and running $PIE^{(n)}$ multiple times can improve the accuracy of disease imaging tracking and reduce the likelihood of missed or misinterpreted changes. Related ablation study results for $M_{ROI}, \gamma, N, \beta_1, \beta_2$ is available in Supplementary E.

**Compare with real longitude medical imaging sequence.** Lack of longitudinal data is a common problem in current chest X-ray datasets. However, due to the spread of COVID, part of the latest released dataset contains limited longitudinal data. In order to validate the disease sequence modeling that PIE can match real disease trajectories, we conduct experiments on generating edema disease progression from 10 patients in BrixIA COVID-19 Dataset (Signoroni et al., 2021). The input image is the day 1 image, and we use PIE to generate future disease progression based on real clinical reports for edema.

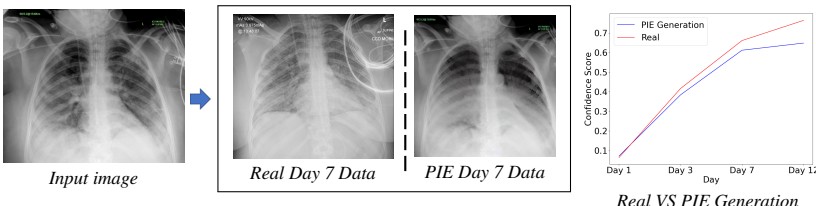

Figure 7: Evaluating the confidence scores of PIE progression trajectories highlights the alignment with realistic progression. The mean absolute error between two trajectories is approximately 0.0658.

**Case study: co-occurring diseases.** PIE is capable of generating images for co-occurring diseases, although the performance slightly trails behind that of single disease generation. To evaluate this ability, we use 10 chest X-ray reports for co-occurring Cardiomegaly, Edema, and Pleural Effusion. 6 cases successfully obtained co-occurring diseases simulation sequence and agreed with experienced clinicians. Figure 8 illustrates an example of disease progression simulation. After 10 steps, all diseases achieve a high confidence score, indicating successful simulation.

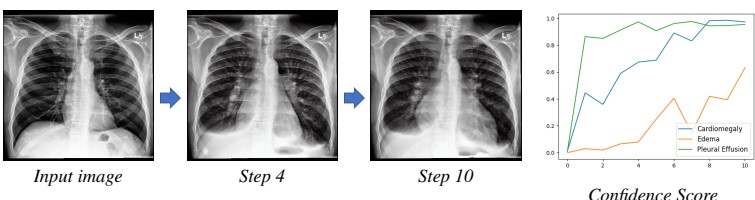

| Input image | Step 4 | Step 10 | Confidence Score |
|---|---|---|---|

Figure 8: PIE can successfully simulate co-occurring disease progression (Patent's clinical report shows high probability to be Cardiomegaly, Edema, Pleural Effusion at the same time).

## 5.4 USER STUDY

To further assess the quality of our generated images, we surveyed 35 physicians and radiologists with 14.4 years of experience on average to answer a questionnaire on chest X-rays. The questionnaire includes disease classifications on the generated and real X-ray images and evaluations of the realism of generated disease progression sequences of Cardiomegaly, Edema, and Pleural Effusion. More details of the questionnaire and the calculation of the statistics are presented in Supplementary F.1. The participating physicians have agreed with a probability of **76.2%** that the simulated progressions on the targeted diseases fit their expectations.

Table 2 provides an interesting insight into experienced physicians' performance in predicting the pathology on real and generated X-rays. Surprisingly, we find users' performance on generated X-rays is superior to their performance on real images, with substantially higher recall and F1. In addition, the statistical test suggests that the F1 scores of generated scans are significantly higher (p-value of 0.0038) than the real scans. One plausible explanation is due to the nature of PIE, the result of running progressive image editing makes pathological features more evident. The aggregated results from the user study demonstrate our framework's ability to simulate disease progression to meet real-world standards.

Table 2: To quantitatively analyze the responses of experienced physicians, we consider each pathology class independent and calculate the precision, recall, and F1 score across all diseases and physicians.

| Data | Precision | Recall | F1 |
|---|---|---|---|
| Real | 0.505 | 0.415 | 0.455 |
| PIE | 0.468 | 0.662 | 0.549 |

## 6 CONCLUSION

In conclusion, our proposed framework, Progressive Image Editing (PIE), holds great potential as a tool for medical research and clinical practice in simulating disease progression. By leveraging recent advancements in text-to-image generative models, PIE achieves high fidelity and personalized disease progression simulations. The theoretical analysis shows that the iterative refining process is equivalent to gradient descent with an exponentially decayed learning rate, and practical experiments on three medical imaging datasets demonstrate that PIE surpasses baseline methods, in several quantitative metrics. Furthermore, a user study conducted with veteran physicians confirms that the simulated disease progressions generated by PIE meet real-world standards. Despite current limitations due to the lack of large amounts of longitude data and detailed medical reports, our framework has vast potential in modeling disease trajectories over time, restoring missing data from previous records, predicting future treatment responses, and improving clinical education. Moving forward, we aim to incorporate more data with richer descriptions and different monitoring modalities, such as chemical biomarkers and physiological recordings, into fine-tuning generative models, enabling our framework to more precise control over disease simulation through text conditioning.

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
