CONTENTS

## A BACKGROUND

**Denoising Diffusion Probabilistic Models** (DDPM) (Ho et al., 2020) are a class of generative models that use a diffusion process to transform a simple initial distribution, such as a Gaussian, into the target data distribution. The model assumes that the data points are generated by iteratively applying a diffusion process to a set of latent variables $x_1, \ldots, x_T$ in a sample space $\chi$. At each time step $t$, Gaussian noise is added to the latent variables $x_t$, and the variables are then transformed back to the original space using a learned invertible transformation function. This process is repeated for a fixed number of steps to generate a final output. The latent variable models can be expressed in the following form,

$$p_\theta(x_0) = \int p_\theta(x_{0:T})dx_{1:T}, \quad where \quad p_\theta(x_{0:T}) \prod_{t=1}^{T} p_\theta^{(t)}(x_{t-1}|x_t) \tag{4}$$

Because of a special property of the forward process,

$$q(x_t|x_0) = \int q(x_{1:t}|x_0)dx_{1:(t-1)} = \mathcal{N}(x_t; \sqrt{\alpha_t}x_0, (1-\alpha_t) \cdot \mathbf{I}) \tag{5}$$

we can express $x_t$ as a linear combination of $x_0$ and a noise variable $\epsilon$, which is the key to enabling the image editing process.

$$x_t = \sqrt{\alpha_t} \cdot x_0 + \sqrt{1-\alpha_t} \cdot \epsilon, \quad where \quad \epsilon \sim \mathcal{N}(\mathbf{0}, \mathbf{I}) \tag{6}$$

**Denoising Diffusion Implicit Models** (DDIM) (Song et al., 2020a) that uses a non-Markovian forward process to generate data. Unlike Denoising Diffusion Probabilistic Models (DDPM), DDIM does not require explicit modeling of the latent variables. Instead, the model generates samples by solving a non-linear differential equation, which defines a continuous-time evolution of the data distribution. We can express its forward process as follows,

$$q_\sigma(x_{1:T}|x_0) := q_\sigma(x_T|x_0) \prod_{t=2}^{T} q_\sigma(x_{t-1}|x_t, x_0) \tag{7}$$

where $q_\sigma(x_{t-1}|x_t, x_0) = \mathcal{N}(\sqrt{\alpha_T}x_0, (1-\alpha_T), \mathbf{I})$ and for all $t > 1$

$$q_\sigma(x_{t-1}|x_t, x_0) = \mathcal{N}(\sqrt{\alpha_{t-1}}x_0 + \sqrt{1-\alpha_{t-1}-\sigma_t^2} \cdot \frac{x_t - \sqrt{\alpha_t}x_0}{\sqrt{1-\alpha_t}}, \sigma_t^2\mathbf{I}) \tag{8}$$

Setting $\sigma_t = 0$, it defines a generation process going from $x_t$ to $x_{t-1}$ as follows

$$x_{t-1} = \sqrt{\alpha_{t-1}}\left(\frac{x_t - \sqrt{1-\alpha_t}\epsilon_\theta^{(t)}(x_t)}{\sqrt{\alpha_t}}\right) + \sqrt{1-\alpha_{t-1}} \cdot \epsilon_\theta^{(t)}(x_t) \tag{9}$$

where the $\epsilon_\theta^{(t)}(x_t)$ is a model that attempts to predict $\epsilon_t \sim \mathcal{N}(\mathbf{0}, \mathbf{I})$ from $x_t$

## B THEORETICAL ANALYSIS

### B.1 PROOF OF PROPOSITION 1

In this proof, we follow the conventions and definitions in Song et al. (2020a)

$$x_{t-1} = \sqrt{\alpha_{t-1}}\left(\frac{x_t - \sqrt{1-\alpha_t}\epsilon_\theta^{(t)}(x_t, y)}{\sqrt{\alpha_t}}\right) + \sqrt{1-\alpha_{t-1}} \cdot \epsilon_\theta^{(t)}(x_t, y) \tag{10}$$

Now given a base image denoted as $\mathbf{x_0^{(0)}}$, we wish to perform diffusion-based editing recursively for $N$ times. The roll-back (to the $k$ th-steps, where $k \geq 1$) according to (6):

$$x_k^{(n)} = \sqrt{\alpha_k} \cdot x_0^{(n-1)} + \sqrt{1-\alpha_k} \cdot \epsilon \tag{11}$$

where $\epsilon \sim \mathcal{N}(0, \mathcal{I})$. Plugging (11) into (10),

$$x_{k-1}^{(n)} = \sqrt{\alpha_{k-1}}\left(\frac{x_k^{(n)} - \sqrt{1-\alpha_k}\epsilon_\theta^{(k)}(x_k^{(n)}, y)}{\sqrt{\alpha_k}}\right) + \sqrt{1-\alpha_{k-1}} \cdot \epsilon_\theta^{(k)}(x_k^{(n)}, y)$$

$$= \sqrt{\alpha_{k-1}} \cdot x_0^{(n-1)} + \sqrt{\frac{\alpha_{k-1}(1-\alpha_k)}{\alpha_k}} \cdot (\epsilon - \epsilon_\theta^{(k)}(x_k^{(n)}, y)) + \sqrt{1-\alpha_{k-1}} \cdot \epsilon_\theta^{(k)}(x_k^{(n)}, y)$$

$$(12)$$

Setting $k = 1$ to perform the revert diffusion process

$$x_0^{(n)} = \sqrt{\alpha_0} \cdot x_0^{(n-1)} + \sqrt{\frac{\alpha_0(1-\alpha_1)}{\alpha_1}} \cdot (\epsilon - \epsilon_\theta^{(1)}(x_1^{(n)}, y)) + \sqrt{1-\alpha_0} \cdot \epsilon_\theta^{(1)}(x_1^{(n)}, y) \qquad (13)$$

Unrolling this recursion in (13), we have

$$x_0^{(N)} = (\sqrt{\alpha_0})^N \cdot x_0^{(0)} + \sqrt{\frac{\alpha_0(1-\alpha_1)}{\alpha_1}} \cdot \sum_i^N (\sqrt{\alpha_0})^i \cdot \epsilon$$

$$+ \frac{\sqrt{\alpha_1 - \alpha_0\alpha_1} - \sqrt{\alpha_0 - \alpha_0\alpha_1}}{\sqrt{\alpha_1}} \cdot \sum_i^N (\sqrt{\alpha_0})^i \cdot \epsilon_\theta^{(i)}(x_1^{(i)}, y)$$

$$(14)$$

Typically, $\alpha_0$ is set to a number close to but less than 1, where in the case of stable diffusion 0.9999, $\alpha_1$ is 0.9995, assuming 50 step schedule.

The second term in (14) is a sampling from Gaussian distributions with geometrically decreasing variances.

$$\lim_{N \to \infty} \sum_i^N (\sqrt{\alpha_0})^i \cdot \epsilon = 0 \qquad (15)$$

Given a large enough N,

$$x_0^{(N)} \approx (\sqrt{\alpha_0})^N \cdot x_0^{(0)} + \frac{\sqrt{\alpha_1 - \alpha_0\alpha_1} - \sqrt{\alpha_0 - \alpha_0\alpha_1}}{\sqrt{\alpha_1}} \cdot \sum_i^N (\sqrt{\alpha_0})^i \cdot \epsilon_\theta^{(i)}(x_1^{(i)}, y) \qquad (16)$$

Proposition 1 in Song et al. (2020a) declares "optimal $\epsilon_\theta^{(i)}$ has an equivalent probability flow ODE corresponding to the "Variance-Exploding" SDE in Song et al. (2020b)". Hence, $\epsilon_\theta^{(i)}(x_1^{(i)}, y) :=$ $\nabla_x \log p(x|y)$. This can be seen as gradient descent with a geometrically decaying learning rate with a factor of $\sqrt{\alpha_0}$, with "base learning rate" $\frac{\sqrt{\alpha_1 - \alpha_0\alpha_1} - \sqrt{\alpha_0 - \alpha_0\alpha_1}}{\sqrt{\alpha_1}}$.

Notice that in (16), there is a decaying factor on the initial image $x_0^{(0)}$, as N grows the original image will surely diminish. Therefore, some other empirical measures are required to preserve the structure of the image, such as segmentation masking, edit strength scheduling and etc.

## B.2 ADDITIONAL THEORETICAL ANALYSIS

### B.2.1 PROOF OF PROPOSITION 2

**Proof** Continuing on 16, for any $n > 1$, we have

$$\|x_0^n - x_0^{n-1}\| = \|(\sqrt{\alpha_0})^n \cdot [(1 - \frac{1}{\sqrt{\alpha_0}}) \cdot x_0^{(0)} - \lambda \cdot \epsilon_\theta^{(n)}(x_1^{(n)}, y)]\|$$

$$\leq (\sqrt{\alpha_0})^n \cdot [\|(1 - \frac{1}{\sqrt{\alpha_0}}) \cdot x_0^{(0)}\| + \lambda \cdot \|\epsilon_\theta^{(n)}(x_1^{(n)}, y)\|] \qquad (17)$$

$$\leq (\sqrt{\alpha_0})^n [(\frac{1}{\sqrt{\alpha_0}} - 1) \cdot C_1 + \lambda \cdot C_2]$$

to guarantee $\|x_0^n - x_0^{n-1}\| \le \delta$, we just need to set,

$$(\sqrt{\alpha_0})^n [(\frac{1}{\sqrt{\alpha_0}} - 1) \cdot C_1 + \lambda \cdot C_2] < \delta$$

$$\frac{n}{2} \log(\alpha_0) + \log((\frac{1}{\sqrt{\alpha_0}} - 1) \cdot C_1 + \lambda \cdot C_2) < log(\delta) \qquad (18)$$

hence, given that $\alpha_0 < 1$ and $\log(\alpha_0) < 0$

$$n > \frac{2}{\log(\alpha_0)} \cdot (log(\delta) - \log((\frac{1}{\sqrt{\alpha_0}} - 1) \cdot C_1 + \lambda \cdot C_2)) \qquad (19)$$

Let $C = \log((\frac{1}{\sqrt{\alpha_0}} - 1) \cdot C_1 + \lambda \cdot C_2)$, we have

$$n > \frac{2}{\log(\alpha_0)} \cdot (log(\delta) - C) \qquad (20)$$

From above, we can conclude that as n grows bigger the changes between steps would grow smaller. The difference between steps will get arbitrarily small.

### B.3    PROOF OF PROPOSITION 3

**Proof**  From 17 and applying triangle inequality, we observe that the difference is a sum of a geometric sequence scaled by a constant factor,

$$\|x_0^{(N)} - x_0^{(0)}\| \le \sum_{n=1}^{N} \|x_0^{(n)} - x_0^{(n-1)}\|$$

$$\le \sum_{n=1}^{N} (\sqrt{\alpha_0})^n [(\frac{1}{\sqrt{\alpha_0}} - 1) \cdot C_1 + \lambda \cdot C_2] \qquad (21)$$

$$= \frac{1 - (\sqrt{\alpha_0})^N}{1 - \sqrt{\alpha_0}} \cdot [(\frac{1}{\sqrt{\alpha_0}} - 1) \cdot C_1 + \lambda \cdot C_2]$$

As $N$ goes to infinity,

$$\lim_{N \to \infty} \|x_0^{(N)} - x_0^{(0)}\| \le \frac{1}{1 - \sqrt{\alpha_0}} \cdot [(\frac{1}{\sqrt{\alpha_0}} - 1) \cdot C_1 + \lambda \cdot C_2] = \kappa \qquad (22)$$

## C    BASELINES

Stable Diffusion Video (SD Video) and Style-Based Manifold Extrapolation (Extrapolation) are two leading techniques in the field of progressive video generation / medical image editing, each displaying promising results within specific domains. However, their applicability remains confined to these particular domains and poses a challenge in extending to the broader scope of different medical imaging data. To illustrate this, Figure 9 provides a comparative visualization of single-step editing using these three techniques.

**Stable Diffusion Video Implementation** (SD Video) (Raw, 2022) control the multi-step denoising process in the Stable Diffusion Videos pipeline. By smoothly and randomly traversing through the sampled latent space, SD Video demonstrates its capability to generate a series of images that progressively align with a given text prompt (see Figure C).

As the state-of-the-art publicly available pipeline, SD Video can generate sequential imaging data by interpolating the latent space via multi-step Stable Diffusion. Though SD Video is useful for general domain (Raw, 2022), it is not controllable for medical prompts.

**Style-Based Manifold Extrapolation Implementation** (Extrapolation)  (Han et al., 2022), involves iteratively modifying images by extrapolating between two latent manifolds. To determine the directions of latent extrapolation, the nearest neighbors algorithm is employed on distributions of

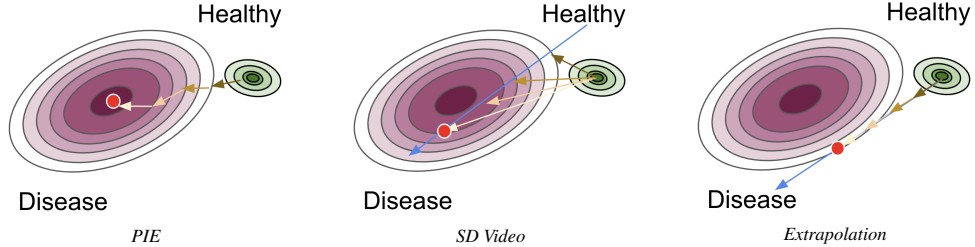

Figure 9: Editing path of PIE, SD Video, and Extrapolation.

known trajectories. However, in cases where progression data is not readily available, as in the study at hand, the directions are obtained by randomly sampling and computing the mean of each manifold.

The actual interpolation of Extrapolation for each step can be defined as:

$$\Delta = \frac{1}{m} \sum_m^{i=1} \frac{\Delta t^i}{\Delta T}[G^{-1}(x_{(0)}^{(n+1)}) - G^{-1}(x_{(0)}^{(n)})] \tag{23}$$

where $G^{-1}(x_{(0)}^{(n)})$ is the corresponding latent vector of the image at stage $n$.

## D EXPERIMENTAL REPRODUCIBILITY

In the following sections, we show the experimental setup used for finetuning the Stable Diffusion model with medical domain-specific medical data. We also outline the design of our disease progression simulation experiment across three datasets, as well as provide an evaluation of the time costs.

### D.1 IMPLEMENTATION DETAILS

Both *PIE* and the baselines use publicly available Stable Diffusion checkpoints (CompVis/stable-diffusion-v1-4) that we further fine-tune on the training sets of each of the target datasets. Our code and checkpoints will be publicly available upon publication.

**Stable Diffusion Training.** To fine-tune the Stable Diffusion model, we center-crop and resize the input images to $512 \times 512$ resolution. We utilize the AdamW optimizer (Loshchilov & Hutter, 2017) with a weight decay set at 0.01. Additionally, we employ a cosine learning rate scheduler (Loshchilov & Hutter, 2016), with the base learning rate set at $5 \times 10^{-5}$. All models undergo fine-tuning for 20,000 steps on eight NVIDIA A100 GPUs, with each GPU handling a batch size of 8.

**DeepAUC DenseNet121 Training.** We have previously outlined the concept of a classification confidence score. In order to pre-train the DeepAUC DenseNet121 model, we utilize the original code repository provided by the authors. The model training process involves the use of an exponential learning rate scheduler, with the base learning rate set to $1 \times 10^{-4}$. Initially, the model is pre-trained on a multi-class task for each respective dataset. Subsequently, we employ the AUC loss proposed in the study by (Yuan et al., 2021) to finetune the binary classification task, distinguishing between negative (healthy) and positive (disease) samples. The AUC loss finetuning process involves the use of an exponential learning rate scheduler, with the base learning rate set to $1 \times 10^{-2}$. All models used for calculating classification confidence score are finetuned for 10 epochs on one NVIDIA A100 GPU, with a batch size of 128. For each class, the final classification accuracy on the validation set is 95.0 % using the finetuned DenseNet121 model.

**PIE progression simulation.** The PIE progression simulation is tested with one NVIDIA A100 GPU. In the following sections, we will show the hyper-parameter search experiment for PIE and explain the insight to adjust them.

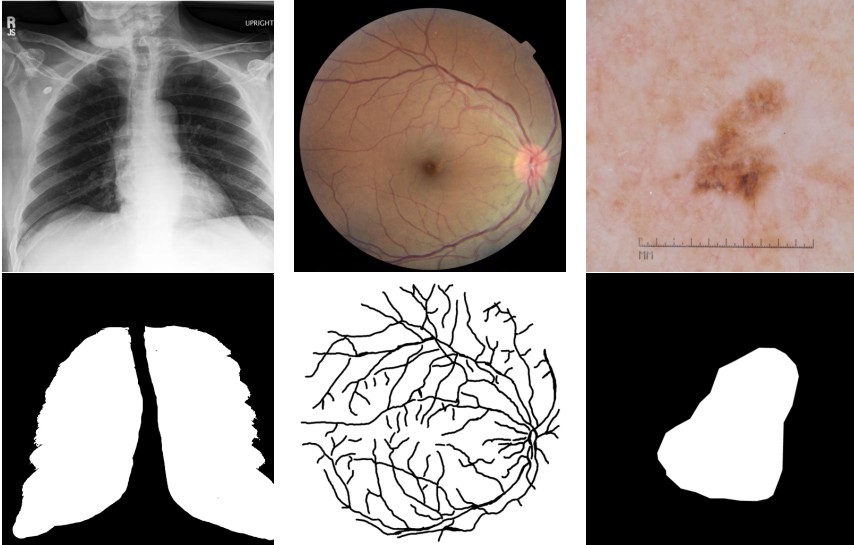

Figure 10: ROI masks for three different domains. Note, the white part in the mask is the disease-related regions.

## D.2 ROI MASK GENERATION

The ROI Masks used in experiments are generated by Segment Anything Model (SAM) Kirillov et al. (2023) and modified to smooth the edge. For different domains, since the region guide prior is different, the mask shape and size are also different. Figure 10 showcases examples of ROI masks utilized for simulation in the PIE and baseline models.

## D.3 TIME COST ANALYSIS

Given that both PIE and the baseline methods utilize the same Stable Diffusion backbone Rombach et al. (2022), a comparison of latency among each method is unnecessary. For simulating disease progression on an image of size $512 \times 512$, per step PIE requires approximately 0.078s to generate the subsequent stage when the strength parameter, $\gamma$, is set to 0.5, batch size is set to 1 and using one NVIDIA A100 (80GB).

## E ABLATION STUDY

To investigate the influence of each hyper-parameter in PIE and analyse the progression visualization, we conduct several ablation studies for visualization, failure cases analysis and hyperparameter searching.

### E.1 VISUALIZATION FOR THREE MEDICAL IMAGING DOMAINS

To provide an in-depth understanding of how PIE performs during different disease progression inference scenarios compared to baseline models, we present detailed visualizations demonstrating PIE's advantages. PIE consistently maintains the realism of the input image even after 10 steps of progression, excelling in most scenarios. Figure 11 displays a comparison among three methods simulating Cardiomegaly progression. PIE outperforms both SD Video and Extrapolation by expanding the heart without introducing noise after 2 steps. Figure 12 displays a comparison among three methods simulating Diabetic Retinopathy progression. PIE outperforms both SD Video and Extrapolation by adding more bleeding (red) and small blind (white) regions without introducing noise after 2 steps. Figure 13 displays a comparison among three methods simulating Melanocytic Nevi progression. PIE outperforms the other two methods as it keeps the color and shape of the patient's skin but continually enhances the feature for Melanocytic Nevi.

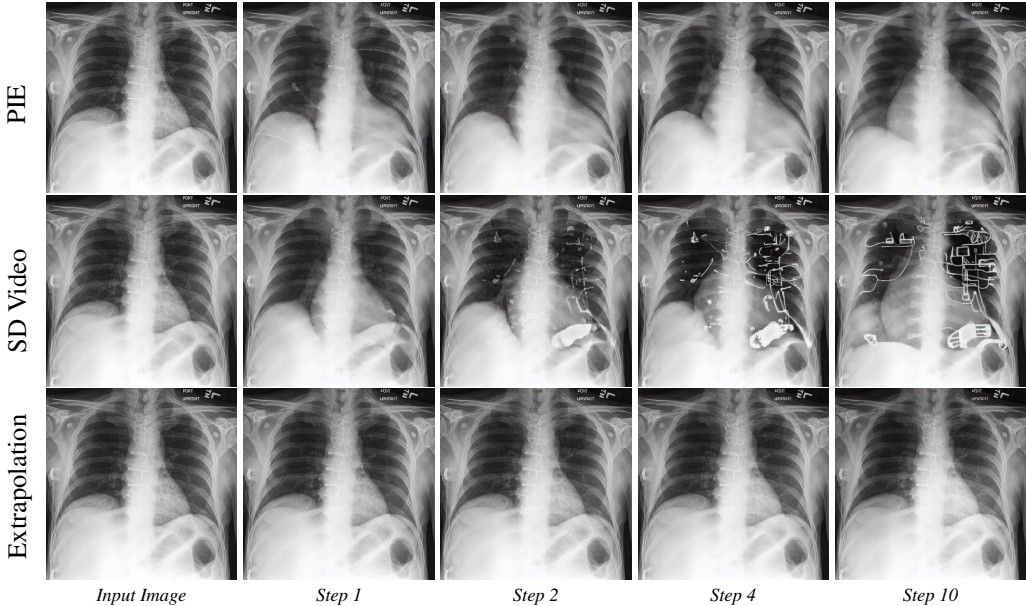

Figure 11: Visualization of PIE, SD Video, Extrapolation to generate disease progression from Cardiomegaly clinical reports.

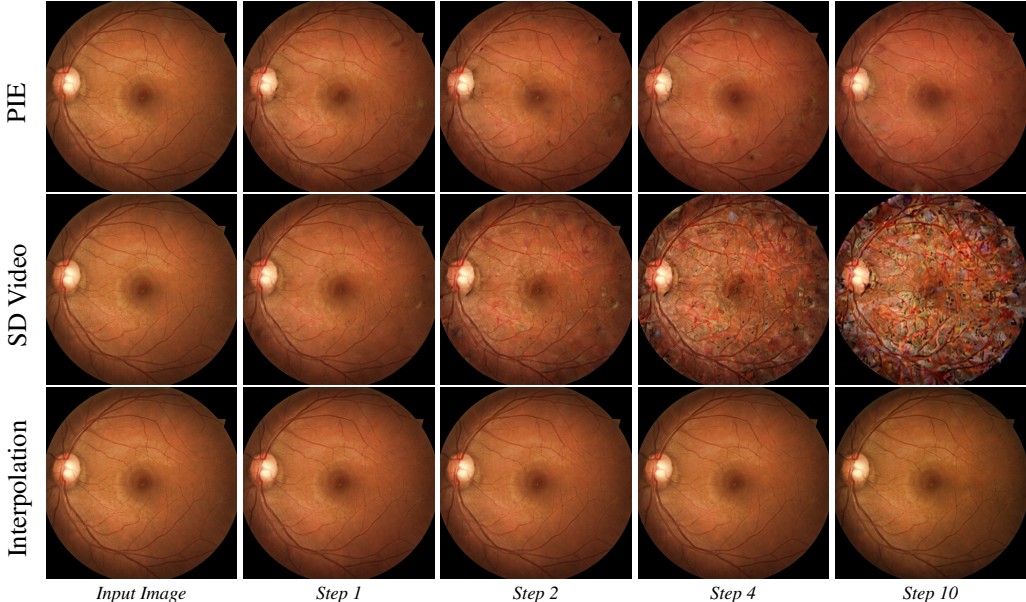

Figure 12: Visualization of PIE, SD Video, Extrapolation to generate disease progression from Diabetic Retinopathy clinical reports.

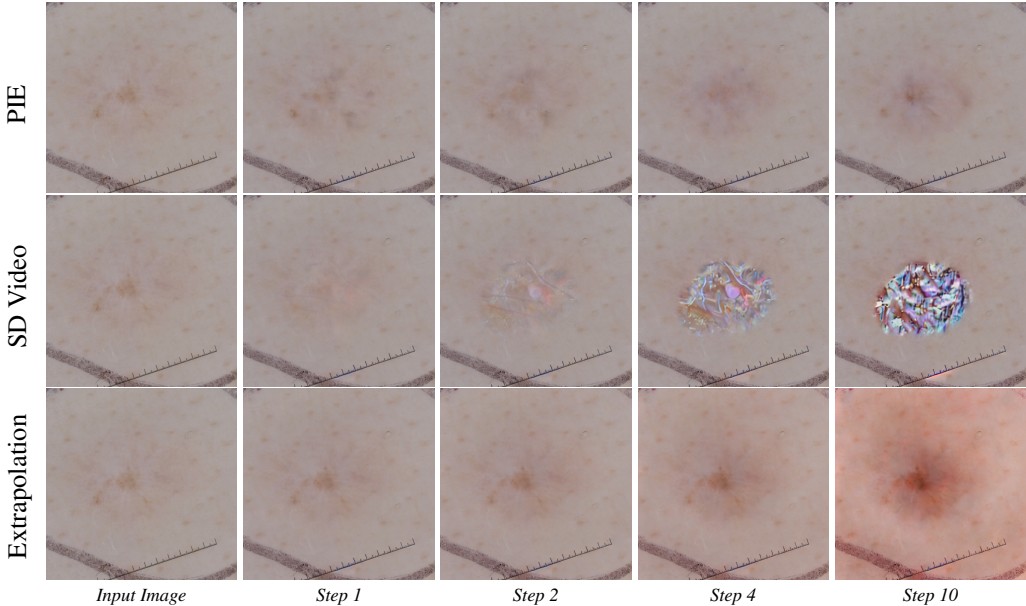

Figure 13: Visualization of PIE, SD Video, Extrapolation to generate disease progression from Melanocytic Nevi clinical reports.

SD Video, while it can interpolate within the prompt latent space, tends to generate noise. Although it can generate convincing video sequences in the general domain, it struggles to simulate authentic disease progression. On the other hand, Extrapolation, despite being effective for bone X-rays, faces challenges with more complex medical imaging domains like chest X-rays. Extrapolation's editing process is considerably slower than the other two methods, and it fails to be controlled effectively by clinical reports.

## E.2 FAILURE CASE ANALYSIS

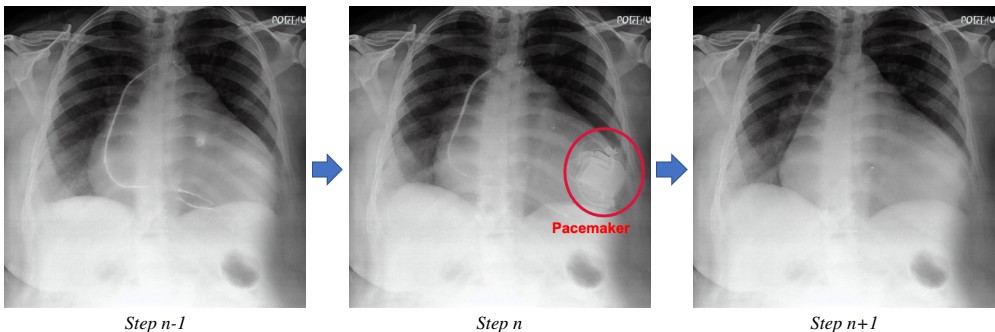

Figure 14: A failure case of the PIE model in preserving the features of a pacemaker during the simulation of Cardiomegaly disease progression. Pacemaker is usually used by patients with severe Cardiomegaly. At Step n-1, the X-ray displays an electronic line. At Step n, both the electronic line and the pacemaker are visible. However, by Step n+1, all the medical device features, including the pacemaker, have vanished from the simulation. It's important to note that the input clinical prompt did not contain any information regarding the pacemaker, making it difficult for the model to retain this crucial feature. This illustrates the challenges faced by models like PIE in dealing with significant but unmentioned clinical features in the input data. It underscores the need for incorporating comprehensive and detailed clinical data to ensure accurate and realistic disease progression simulations.

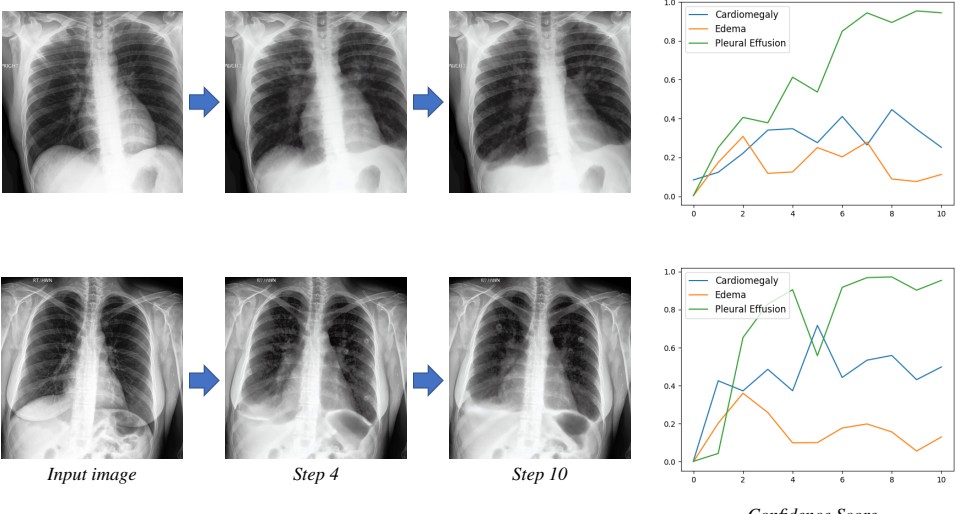

Figure 15: Two failure cases of the PIE model in simulating co-occurring diseases progression for Cardiomegaly, Edema, and Pleural Effusion, only the features for Pleural Effusion are captured. These failure cases arise from the issue related to imbalanced label distribution in the training data. Specifically, the prevalence of Pleural Effusion is significantly higher than the other four classes, leading to an inherent bias in the model's simulations for co-occurring diseases. This imbalance emphasizes the need for a more diversified and balanced training dataset for more accurate simulation of co-occurring diseases.

Despite outperforming baseline models, PIE still faces limitations tied to data sensitivity issues. For instance, imbalances in the distribution of training data for Stable Diffusion can limit PIE's capability to edit rare diseases. In some cases, PIE might generate essential medical device features but fail to preserve them in subsequent stages of progression simulation, as observed with features like pacemakers. Figure 14 shows a good example of pacemaker disappearance during simulation. Besides, the co-occurring diseases simulation also has some failure cases (see Figure 15).

Fundamentally, these shortcomings could be addressed with a larger and label-equal distribution dataset. However, given that the volume of medical data is often smaller than in other domains, it is also important to explore fine-tuning PIE's diffusion backbone through few-shot learning under extremely imbalanced label distribution.

### E.3 HYPERPARAMETER SEARCH & ANALYSIS

While the Fréchet Inception Distance (FID) and Kernel Inception Distance (KID) metrics are widely utilized in the evaluation of generative models, they do not necessarily align with the perceptual quality of images, making them unsuitable for evaluating progression simulation tasks. Nevertheless, for the subsequent hyperparameter search and ablation study, we added the results of KID score, given its ability to provide insight into the diversity and distributional closeness of the generated images in relation to the real data.

To demonstrate the significance of the ROI mask as a key control factor in the PIE, an experiment was conducted to compare the performance of models using and not using ROI masks across three domains, with each model tested using 5 different random seeds. The evaluation focused on the classification confidence score and CLIP score. Results, as shown in Table 3, revealed that while removing the ROI mask could sometimes increase the confidence score (observed for chest X-ray and skin imaging), it consistently led to a decrease in the CLIP score. Further visual analysis depicted in Figure 16 shows that the ROI mask crucially helps in preserving the basic shape of the medical imaging during the PIE process. Consequently, these findings suggest that the ROI mask, alongside clinical reports, serves as a critical medical prior for simulating disease progression. It helps the PIE to concentrate on disease-related regions while maintaining the realism of the input image.

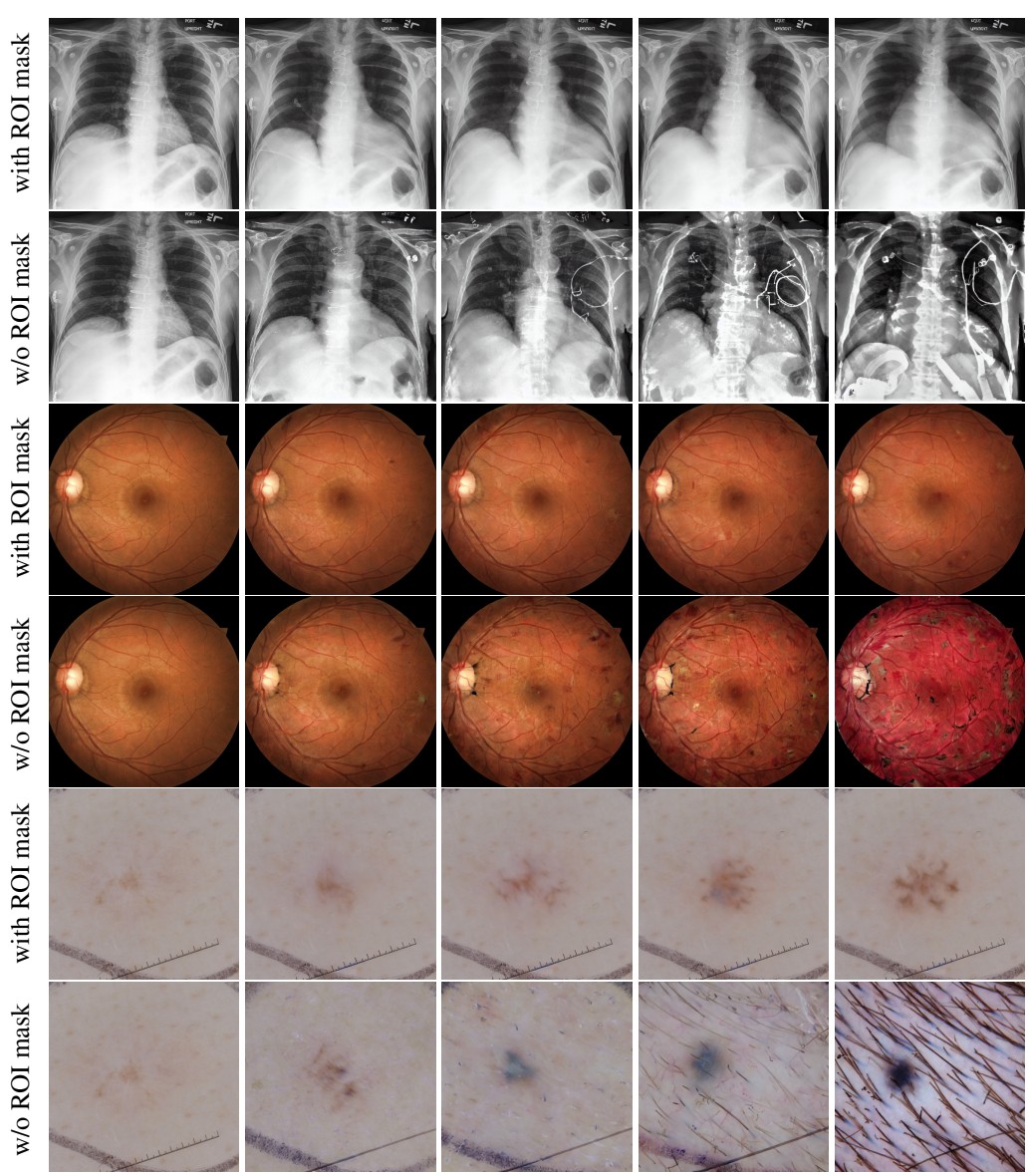

Figure 16: Visualization comparison between ROI mask influence for three medical imaging domains.

We present a comprehensive examination of various hyperparameters, namely strength ($\gamma$ in Algorithm 1), step ($N$), $\beta_1$, and $\beta_2$, and their respective tradeoffs as demonstrated in tables 4, 5, and 6. Notably, a discernible tradeoff exists between classification confidence score and CLIP-I/KID, wherein an increase in classification confidence score results in a decrease in CLIP-I. In the subsequent section, we offer an intricate analysis of each hyperparameter.

Table 4 illustrates a positive correlation between the strength of PIE and classification confidence score, while revealing a negative relationship between strength and CLIP-I/KID. From an intuitive standpoint, as the strength of PIE increases, more features are directed to align with the pathologies within the original images. Consequently, the classifier exhibits greater confidence in accurately predicting the specific disease class, leading to a more significant deviation from the initial starting point. As a result, the classification confidence score value increases while the CLIP-I value decreases, reflecting the inverse relationship between the two metrics.

Table 5 presents a similar pattern initially. However, both classification confidence score and CLIP-I/KID reach a state of stability after a certain number of steps. This observation aligns with our theoretical analysis as outlined in Proposition 2. Moreover, it can be interpreted as the convergence of a Cauchy geometric sequence, wherein the discrepancy between successive steps gradually diminishes as the value of $N$ tends towards infinity.

Lastly, in Table 6, we explore the interplay between $\beta_1$ and $\beta_2$, which serve as parameters governing the rate of progression within and outside the region of interest (ROI) respectively. Our findings reveal that $\beta_2$, responsible for regulating the pace of progression within the ROI, exerts a more pronounced influence on classification confidence score, while $\beta_1$ exhibits a stronger impact on CLIP-I/KID. This outcome can be intuitively comprehended, considering that the ROI typically encompasses a smaller area of paramount importance for aligning with disease-specific features. Conversely, the areas outside the ROI exert a significantly greater influence on the realism captured by CLIP-I/KID.

In summary, our study elucidates the inherent trade-offs and offers valuable practical insights, thereby furnishing meaningful guidance for effectively utilizing PIE in practice.

Table 3: Mask, w/o mask guidance comparisons.

| Method | Chest X-ray | | Retinopathy | | Skin Lesion Image | |
|---|---|---|---|---|---|---|
| | Conf ($\uparrow$) | CLIP-I ($\uparrow$) | Conf ($\uparrow$) | CLIP-I ($\uparrow$) | Conf ($\uparrow$) | CLIP-I ($\uparrow$) |
| w/o mask | 0.729 | 0.93 | 0.163 | 0.96 | 0.666 | 0.85 |
| with mask | 0.690 | 0.96 | 0.807 | 0.99 | 0.453 | 0.95 |

Table 4: Strength $\gamma$ selection for $N = 10$.

| Strength | Conf ($\uparrow$) | CLIP-I ($\uparrow$) | KID ($\downarrow$) |
|---|---|---|---|
| 0.05 | 0.038 | 0.966 | 0.0685 |
| 0.1 | 0.120 | 0.969 | 0.0638 |
| 0.2 | 0.273 | 0.969 | 0.0885 |
| 0.3 | 0.455 | 0.967 | 0.1033 |
| 0.4 | 0.746 | 0.965 | 0.1142 |
| 0.5 | 0.977 | 0.962 | 0.1389 |
| 0.6 | 0.995 | 0.956 | 0.1549 |
| 0.7 | 0.998 | 0.955 | 0.1533 |
| 0.8 | 0.999 | 0.951 | 0.1629 |

# F    USER STUDY

## F.1    QUESTIONNAIRE

The questionnaire in the survey is approved by Affiliated clinical research institute. The questionnaire includes 2 parts. Part one consists of 20 multiple choices of single image classifications, 10 single-step

Table 5: Simulation steps $N$ selection with $\gamma = 0.5$.

| Step ($N$) | Conf ($\uparrow$) | CLIP-I ($\uparrow$) | KID ($\downarrow$) |
|---|---|---|---|
| 1 | 0.491 | 0.965 | 0.094 |
| 2 | 0.731 | 0.964 | 0.098 |
| 5 | 0.881 | 0.963 | 0.121 |
| 10 | 0.978 | 0.962 | 0.142 |
| 20 | 0.989 | 0.961 | 0.111 |
| 50 | 0.975 | 0.962 | 0.130 |
| 100 | 0.959 | 0.962 | 0.115 |

Table 6: Beta selection.

| $\beta_1$ | $\beta_2$ | Conf ($\uparrow$) | CLIP-I ($\uparrow$) | KID ($\downarrow$) |
|---|---|---|---|---|
| 0.01 | 1.0 | 0.954 | 0.946 | 0.133 |
| 0.01 | 0.75 | 0.977 | 0.948 | 0.140 |
| 0.01 | 0.5 | 0.554 | 0.965 | 0.090 |
| 0.1 | 1.0 | 0.960 | 0.965 | 0.126 |
| 0.1 | 0.75 | 0.976 | 0.962 | 0.140 |
| 0.1 | 0.5 | 0.554 | 0.962 | 0.089 |
| 0.2 | 1.0 | 0.963 | 0.947 | 0.134 |
| 0.2 | 0.75 | 0.977 | 0.964 | 0.137 |
| 0.2 | 0.5 | 0.556 | 0.962 | 0.089 |

generations, and 10 real X-ray images sampled from the training set. Part two consists of 3 generated disease progressions consisting of Cardiomegaly, Edema as well Pleural Effusion. Each progression runs 10 steps. For each single image classification, we ask "Please determine the pathologies of the following patient" and let the user pick from 6 options {No findings, Cardiomegaly, Consolidation, Edema, Effusion, Atelectasis} with possible co-occurrence, while for each of the 10-step progressions, we ask "Does the below disease progression fit your expectation?" and let the user input a binary answer of yes or no.

Below we include the full instructions we gave for our user study both in English and Chinese. Here we only include the English version of questions. The examples are shown in Figure 19 and 20.

1. Please read the instructions and inspect the images carefully before answering.

2. Please provide your years of experience

3. For the first 20 questions, please determine the pathologies from the X-ray images (you can choose more than one answer). For the last 3 questions, please answer if the disease progression shown fits your expectations.

## F.2 STATISTICS

The distribution of the years of experience from the group of physicians who participated in the user study. The average number of years of experience is $14.4$ years. Over half of the group have more than 10 years of experience. This data attests that our surveyees are highly professional and experienced.

To show the significance of our findings, we also performed the paired t-test on the F1 scores of real and generated scans over the 35 users. Our finding is significant with a p-value of $0.0038$.

To quantitatively analyze their responses, we treat each class of pathologies as an independent class and compute precision, recall, and F1 over all the pathologies and physicians.

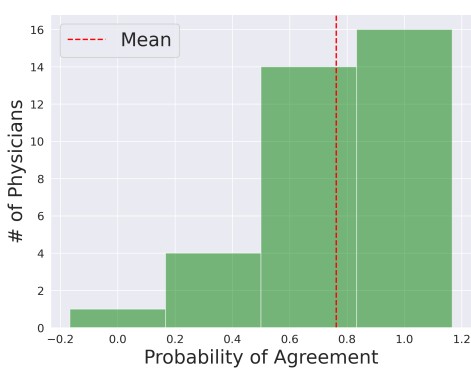

Figure 17: Distribution of probability of agreement to the generated progressions among 35 veteran physicians.

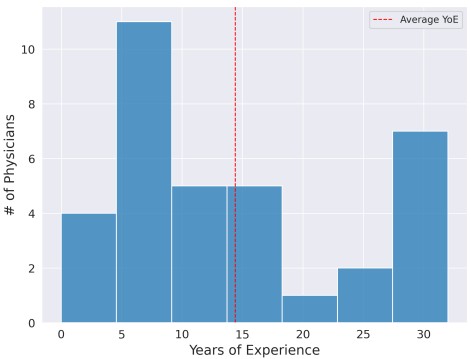

Figure 18: The distribution of the years of experience from the group of physicians who participated in the user study. The average number of years of experience is $14.4$ years.

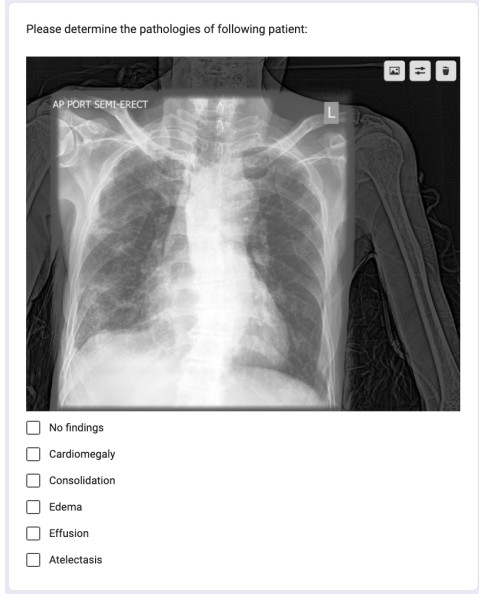

Figure 19: Example of User Study I: we ask the physicians to pick from the 6 options and it's possible to pick more than one option.

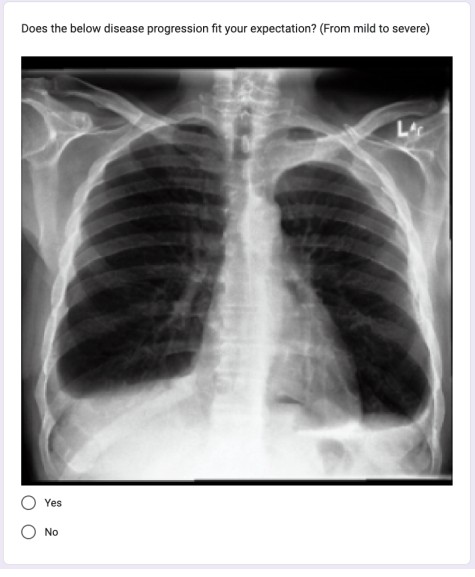

Figure 20: Example of User Study II: we ask the physicians to decide if the generated progression of the disease is credible or not.

## G DISCUSSION AND ETHICS STATEMENT

The proposed framework is subject to several limitations, with one of the primary constraints being the limited scope of Stable Diffusion. Due to the model's pre-training on general domain data, the absence of detailed medical reports poses a significant challenge to the model's ability to accurately and reliably edit medical images based on precise text conditioning. Moreover, the framework's overall performance may be influenced by the quality and quantity of available data, which can limit the model's accuracy and generalizability. Furthermore, the absence of surgical or drug intervention data further restricts the framework's ability to simulate medical interventions. Moving forward, it would be beneficial to explore ways of integrating more detailed descriptions of medical data in the fine-tuning process of Stable Diffusion to improve the framework's performance and precision in disease simulation through text conditioning. Additional details on the framework's limitations, including an analysis of failure cases, can be found in Supplementary E.

Progressive Image Editing (PIE) holds promise as a technology for simulating disease progression, but it also raises concerns regarding potential negative social impacts. One crucial concern revolves around the ethical use of medical imaging data, which may give rise to privacy and security issues. To address this, healthcare providers must take measures to safeguard patient privacy and data security when utilizing PIE. An effective mitigation strategy involves employing anonymized or de-identified medical imaging data, while also adhering to ethical guidelines and regulations like those outlined by HIPAA (Health Insurance Portability and Accountability Act).

Another concern relates to the accuracy of the simulations generated by the framework, as errors could lead to misdiagnosis or incorrect treatment decisions. To alleviate this concern, rigorous testing and evaluation of the technology must be conducted before its implementation in a clinical setting. Enhancing the accuracy of simulations can be achieved by incorporating additional data sources, such as patient history, clinical notes, and laboratory test results. Additionally, healthcare providers should receive adequate training on effectively utilizing the technology and interpreting its results.

Discrimination against certain groups, based on factors such as race, gender, or age, poses yet another potential concern. Healthcare providers must ensure the fair and unbiased use of the technology. This can be accomplished by integrating diversity and inclusion considerations into the technology's development and training processes, as well as regular monitoring and auditing its usage to identify any signs of bias or discrimination.

The cost and accessibility of the technology present further concerns, potentially restricting its availability to specific groups or geographic regions. To tackle this issue, healthcare providers should strive to make the technology accessible and affordable to all patients, regardless of socioeconomic status or geographic location. This can be achieved through the creation of cost-effective models, partnerships with healthcare providers, and government funding initiatives.

Lastly, there is a risk of excessive reliance on technology, leading to a diminished reliance on clinical judgment and expertise. To mitigate this concern, healthcare providers must be trained to view the technology as a tool that complements their clinical judgment and expertise, rather than relying solely on it for diagnostic or treatment decisions. The technology should be used in conjunction with other data sources and clinical expertise to ensure a comprehensive understanding of disease progression.

In conclusion, while the use of PIE comes with potential negative social impacts, there are viable mitigations that can address these concerns. Healthcare providers must be aware of the ethical implications associated with this technology and take appropriate measures to ensure its safe and responsible utilization.