# OpenReview forum: "PIE: Simulating Disease Progression via Progressive Image Editing"
_ICLR.cc/2024/Conference — Submitted to ICLR 2024_

### Official Review · Reviewer_GQCh · 2023-10-25

**Soundness:** 3 good
**Presentation:** 3 good
**Contribution:** 2 fair
**Rating:** 5
**Confidence:** 4

**Summary:**

The paper proposes a generative model based on DDIM called PIE (progressive image editing) to simulating disease progression using a text-conditioned stable diffusion model. The theoretical proofs show how the changes to the image being edited is bounded by a constant and converges. The approach is benchmarked on datasets involving lung x-rays, diabetic retinopathy, and skin lesions and shows promising performance against 2 other baselines, namely stable diffusion video and style based manifold extrapolation. The models are evaluated based on CLIP scores and classification confidence scores on the generated images. In addition to these experiments, a real-world edema progression and a user study is also shown to provide evidence that the disease progression makes sense.

**Strengths:**

1. The work is focussed on an important problem if simulating missing longitudinal data in medical imaging. Scarcity of such data is a genuine issue and to my knowledge, this work is among the few which attempts to do so without access to any temporal image data.
2. The proposed approach performs well on confidence score metrics and shows gradual improvements with number increasing of steps. The use of the real-world dataset which sequential images and the user study provide good evidence of the efficacy of the approach. Similarly, showing performance on 3 datasets from different problem types is also a big plus.
3. The availability of the code and the details in the supplementary are appreciated and a strong sign towards transparency and reproducibility. The experiments on ablations and sensitivity to hyperparamaters is also helpful for trying this approach and for future extensions of this work.

**Weaknesses:**

1. Even though the paper shows the editing process to be bounded and converging, I find it hard to understand why generating disease progression in images without any intermediate temporal information will lead to the correct intermediate pathologies in the image. Neither the text, nor the image have any information about what temporally intermediate stages of the disease can look like. Lacking this info, it's not clear how the progression is constrained to be realistic or biologically plausible. The real-world experiments on the edema dataset as well as the user study are most certainly helpful, but not completely convincing.
2. The paper proposes an interesting solution to a medical imaging problem, but is technically incremental in terms of the proposed method since it's a direct application of DDIM for conditional generation.
3. The performance improvement with PIE is less significant on CLIP metrics. Additionally, all the similarity numbers on all datasets and baselines are usually high (>0.9 for a metric having a range of [-1,1]) which perhaps points to the fact that differences in this metric might not be hugely indicative of better fidelity, specially for medical images.

**Questions:**

1. Why are the confidence scores for other baselines so bad for the diabetic retinopathy dataset?
2. The recall, and in turn F1 scores for the simulated images is higher than the real ones in the case study. If this is indeed due to the simulated images accentuating the disease features, does that pose as a risk to this technique, specially in situations where it hallucinates or exaggerates pathologies?
3. Not a question, but the presence of RoI masks seems very important as without them, the model hallucinates significantly (as shown in the supplementary). It might be worth including this in the limitations sections or making this explicit for the readers if not already done so.

---

> ### Author Response · Authors · 2023-11-21
> **Response to Reviewer GQCh**
>
> Weaknesses 1's reply: One insight of PIE is even if it is difficult to obtain real disease progression from the same patient, it is still possible to use all data across different patients to build an unified model to predict disease. Simulating disease progression in images without direct temporal data relies heavily on the learned distribution of generative models, the quality and diversity of the training data, domain knowledge, and iterative refinement processes.
> Generative models like diffusion models are often used to simulate realistic images. These models can learn complex distributions from a dataset and generate new instances that are statistically similar to the observed data. In PIE, text acts as a locator of which part of implicit distribution the inference step would use. Even without temporal data, diffusion models can still learn correlations between disease stages and specific visual features.
>
> Weaknesses 2's reply: We propose a framework to solve the medical imaging problem with plugged in generative models. DDIM happens to have certain nice properties of convergence as well as state of the art generation quality on images.In addition, it’s also easier to train and adapt given our limited resources. We could technically swap in other text-image generative models, as long as they share the same convergence properties as DDIM.
>
> Weaknesses 3's reply: Actually, data visualization is better than CLIP-I for fidelity evaluation. This is because medical images have a domain gap to the general domain. As shown qualitatively in figure 4 and figures in appendix, images produced by earlier steps tend to be closer to real image distribution, which happens across all three methods. This tends to average out our advantage in CLIP-I scores. However, the quality of the generated images would evidently show that the improvement in CLIP-I scores are definitely NOT marginal.
>
> Question 1's reply: Diabetic retinopathy dataset is the easier one in the selected datasets. CheXpert contains over 100K imaging data for 6 classification and containing co-occurrence diseases. Kaggle Diabetic Retinopathy Detection Challenge contains over 40K imaging data for 2 classification and ISIC 2018 contains about 10K data for 4 classification. When datasets contain more co-occurrence disease cases and less data, it will be difficult to pretrain generative models.
>
> Question 2's reply: Yes, as with all generative models, it’s pruned to hallucinate, but we believe with larger and better curated datasets, this problem can be overcome. We hope this paper can serve as the starting point and provide the community the motivation and incentive to get the ball rolling on disease progression simulations.
>
>
> Question 3's reply: Thanks for pointing out. We will move the ROI mask ablation study to the main body.

---

### Official Review · Reviewer_KqFr · 2023-10-31

**Soundness:** 2 fair
**Presentation:** 3 good
**Contribution:** 3 good
**Rating:** 5
**Confidence:** 4

**Summary:**

The paper proposes a method to generate realistic medical images corresponding to progression of diseases. The input is the image to be progressed and a text prompt describing the progression in the form of a clinical report. The method using Denoising Diffusion Implicit Models (DDIM) and text encoding using CLIP. It is evaluated on a dataset of chest X-rays (CheXpert), and skin cancer (ISIC 2018/HAM10000), and Diabetic Retinopathy. The approach is compared to Stable Diffusion Video and Style-Based Manifold Extrapolation. The results are evaluated qualitatively using visual examples and quantitatively by comparing CLIP embeddings of real and generated images and using the confidence score of a disease classifier. In addition 35 physicians and radiologists were surveyed using a questionaire on the realism of the generated images.

**Strengths:**

- The ability to simulate disease progression in medical images could have many relevant uses.

- Evaluated on a number of different medical imaging modalities.

- The results seem to be of good quality and the method novel.

- Trained model checkpoints will be made available on publication according to the supplement.

**Weaknesses:**

- A fundamental problem with the work is the focus and claims related to modelling of disease trajectories or progression. It is not entirely clear what the authors mean when they use these terms, and since this is a critical part of the work, this should really be defined.  Disease trajectory, I would understand to refer to the course of a disease over time. This could be in an individual or maybe as an average in a population. This would imply some predictive capability, and we are also told this in the abstract (see below). Yet there is as far as I can see no evidence that the proposed method can predict the future of individual patients or average patients. Instead it seems to me that what the approach is doing is instead to create images corresponding to different disease severities, which is certainly interesting, but a very different and generally easier problem. Loosely described, this could perhaps be called disease progression simulation, which is also a term used by the manuscript in places.

- "PIE can allow healthcare providers to model disease imaging trajectories over time, predict future treatment responses" - where is the evidence for this?

- "Specifically, we leverage recent advancements in text-to-image generative models to simulate disease progression accurately and personalize it for each patient." - how is it personalized?


- "The learning rate in this iterative process is decaying exponentially with each iteration forward, which means that the algorithm is effectively exploring the solution space while maintaining a balance between convergence speed and stability.", I don't think this is supported by evidence/references.

- "The physicians agree that simulated disease progressions generated by PIE closely matched physicians’ expectations 76.2% of the time, indicating high accuracy and quality." - is this a relevant measure to compare to? Are physicians able to predict actual progression?

- The question the physicians were asked appears to be "Does the below disease progression fit your expectation?" It is unclear if this is supposed to match a development in disease severity or what the specific development in this particular case would be expected to be.

- "However, all these methods have to use full sequential images and fail to address personalized healthcare in the imaging space. The lack of such time-series data, in reality, poses a significant challenge for disease progression simulation". I am uncertain about what is meant by "failing to address personalized healthcare in the imaging space". Could more precise wording be used? Also I feel like the authors are overly focused on the requirement of sequential data as a limitation. Longitudinal data exists for a reason and it may be much more difficult if not impossible to derive individualized progression models from cross-sectional data alone. I would suggest the authors think about the wording here and present it not as a limitation of previous methods but rather as a situation where the proposed approach could be used where previous models may not.

- Explain abbreviation DDIM

- "Due to the properties of DDIM, the step size would gradually decrease
with a constant factor.", what step size? No mention of step size before this point.

- Proposition 2 and 3, would benefit from some motivation, and explanation in text. There are variables and functions used without definition.

- "In addition, Proposition 2 and 3 show as n grows bigger, the changes between steps would grow smaller. Eventually, the difference between steps will get arbitrarily small. Hence, the convergence of P IE is guaranteed and modifications to any inputs are bounded by a constant." - I don't see how this follows. Could you help the reader a bit?

- What are the numbers presented in Table 1?

- "To further assess the quality of our generated images, we surveyed 35 physicians and radiologists with 14.4 years of experience on average to answer a questionnaire on chest X-rays." - why are the questions asked not

- "Furthermore, a user study conducted with veteran physicians confirms that the simulated disease progressions generated by PIE meet real-world standards.", what real world standards?

**Questions:**

- See the fundamental weakness mentioned in the above. Is it the authors intention to claim that the method can be used for prediction of future time points?

---

> ### Author Response · Authors · 2023-11-21
> **Response to Reviewer KqFr**
>
> Weaknesses 1's reply: we conduct experiments to predict the potential disease future of individual patients. In the ablation study, we compare PIE generated results with real longitude medical imaging sequences. In order to validate the disease sequence modeling that PIE can match real disease trajectories, we conduct experiments on generating edema disease
> progression from 10 patients in BrixIA COVID-19 Dataset (Signoroni et al., 2021). The input image is the day 1 image, and we use PIE to generate future disease progression based on real day 1 clinical reports for edema. The 10 generated and the real data pairs from Figure.7 have been checked by clinical doctors with 80% agreement.
>
> Weaknesses 2's reply: Thanks for pointing out this misunderstanding sentence. For CheXpert, EHR (including future treatment) are shown in part of the report. For ISIC 2018, Kaggle Diabetic Retinopathy Detection Challenge, the report doesn't contain therapy information. We will rewrite this sentence to show the author only part of datasets can predict future treatment responses in image space.
>
> Weaknesses 3's reply: It can be personalized by using the patients’ medical records as text instruction and an initial medical image as input for PIE.
>
> Weaknesses 4's reply: We would argue otherwise, our experiment results show that in the end it can generate realistic images with verifiable disease features without diverging to human unreadable noise like other methods do. We believe that itself is the best evidence. Our ablation study in the appendix also shows that our method converges in around 10 steps, which is fast and practical number of steps to solve the problem.
>
> Weaknesses 5's reply: A well-experienced radiologist can predict disease progression based on their understanding of CT, X-ray, MRI imaging. However, they will not make deterministic statements.
>
> Weaknesses 6's reply: The reason is: the input patient's X-ray image and clinical report is a state that has already been diagnosed with that disease, but not achieved a serious condition. Thus, we used a proposed model to simulate the potential changing sequence. We also conducted comparative experiments using real longitudinal data, and the results showed that the confidence score of the PIE-generated image and the judgments of the three clinicians were close to the real progress. Previous work must use large amounts of longitudinal data, which are, however, almost impossible to obtain.
>
> Weaknesses 7's reply: Thanks for your suggestion. We will adjust the wording accordingly.
> “personalized healthcare in the imaging space” means the current datasets don't contain each patient’s long-term history data (MRI, X-ray, CT,...).
>
> Weaknesses 8's reply: Denoising Diffusion Implicit Models.
>
> Weaknesses 9's reply: We discussed the details of proposition 1 in the appendix section B1, including the definition of “learning rate”. We show that $\epsilon^{(i)}\_{\theta}(x_1^{{(i)}}, y) := \nabla_{x} \log p(x | y)$. This can be seen as gradient descent with a geometrically decaying learning rate with a factor of $\sqrt{\alpha_0}$, with "base learning rate" $\frac{\sqrt{\alpha_1 - \alpha_0\alpha_1} - \sqrt{\alpha_0 - \alpha_0\alpha_1}}{\sqrt{\alpha_{1}}}$.
>
> Weaknesses 10's reply: The purpose of proposition 2 and 3 is to show PIE will lead to a stable convergence, as we claimed in the paper: “the changes between steps would grow smaller”. We apologize if variables are not self-explanatory enough in the propositions, we will add more text for clarifications in the later versions.
>
> Weaknesses 11's reply: As stated in the assumption of proposition, we since we assume the denoising step generated is bounded by a constant. Hence the only thing determining the scale of changes is the step size, which is very similar to “learning rate” in SGD.
>
> Weaknesses 12's reply: They are the confidence scores and the CLIP-I scores which measure how good the generated images are based on the ground truth.
>
> Weaknesses 13's reply: Half of the medical doctors and radiologists only finished the survey and did not provide the reason. Here are some key questions from the other doctors who feel strange about the progression: (1) no co-occurrence disease in the image; (2) pacemaker disappear. We have discussed these issues in the appendix.
>
> Weaknesses 14's reply: The real world standards mean the real disease progression verified by multiple experienced medical doctors. This might not be the golden standard but that is the best we can get. In the future, it can be improved by more clinical professionals.
>
> For Question: We would like to express our sincere gratitude to Reviewer KqFr for pointing out these weaknesses and giving us potential improvement directions. Yes, this method is used for the disease prediction for future time points and we have proven it can be useful to predict Edema in existing COVID dataset.

---

> > ### Comment · Reviewer_KqFr · 2023-11-21
> > **Thanks but...**
> >
> > Thank you for a very detailed response. As a reviewer, I'm looking for more than just your thoughts on my comments or questions. It's important to directly address how you plan to revise your paper in response to my comments, if at all. It is fine to leave it as is, but I would appreciate your thoughts on why at least.
> >
> > While I think the approach is interesting, I think it is a too strong claim that the method can predict future time-points. I would expect much more solid evidence for this. 10 images is a starting point but too limited to say anything with much confidence. Moreover is 80% agreement clinically relevant? I think there is a lot of work to be done before we can confidently state that the approach is able to predict individual patient's future.

---

### Official Review · Reviewer_rTQB · 2023-10-31

**Soundness:** 3 good
**Presentation:** 2 fair
**Contribution:** 1 poor
**Rating:** 3
**Confidence:** 3

**Summary:**

The manuscript presents a framework for progressively editing a medical image to simulate disease progression. The method is based on a diffusion denoising model that generates medical images based on text (medical report). The method is showcased in three medical applications to simulate enlarged disease sites and more server disease effects.

**Strengths:**

1. The task of editing medical images to inject or remove disease effects is of interest and is related to a long-standing problem of counter-factual generation.

2. The model generates visually authentic disease effects that are better than two comparison baselines.

**Weaknesses:**

1. Methodologically, the text condition seems to be a major part of the proposal. In fact, I believe it is the only mechanism that allows to model to "know" what is a "disease effect". However, it is discussed minimally in the method section, and is never discussed experimentally.

2. A core in these generative models in medical imaging is to show that the model does not hallucinate; the generated subject-specific disease should reflect realistic progression. The paper lacks quantitative evaluation on this aspect. The only experiment (Fig. 7) shows that the simulated disease effect deviates largely from the real case.

3. I'm having a hard time imagining what would be an ideal use scenario. The manuscript argues that the method can be used for "model disease imaging trajectories over time, predict future treatment responses, fill in missing imaging data in clinical records, and improve medical education". I'm not convinced it can do all of those things except for the last goal of "medical education", where the method can generate synthetic disease effects without showing an actual patient's data (see my questions below)

**Questions:**

1. It seems that the model cannot generate a deterministic progression trajectory as it mentions "We obtain at least 50 disease imaging trajectories for each patient". Why is this desired? How can such randomness contribute to "model disease imaging trajectories over time, predict future treatment responses, fill in missing imaging data in clinical records"?

2. I'm not sure why the model should generate "disease effects" from a healthy image (e.g. Fig. 5 3rd row). Isn't this contradictory to "predict future treatment responses" or "model disease imaging trajectories"? Healthy subjects should simply have healthy trajectories.

---

> ### Author Response · Authors · 2023-11-21
> **Response to Reviewer rTQB**
>
> Weaknesses 1's reply: We thank you for your suggestion. However, text condition is not the major part of the proposal as it is the basic structure of any stable diffusion-based generative models such as DreamBooth and ControlNet for image generation. These articles generally do not conduct ablation experiments on text conditions as these text inputs are encoded with frozen CLIP text encoder. As you suggested, we conducted an extension experiment based on testing different text conditions in the MIMIC-CXR dataset for PIE's simulation results. For experiment, we only use the test set and the simulation steps are 10 for each disease sequence.
>
> | Clinical context | Source | Confidence score |
> | --------------- | --------------- | --------------- |
> MIMIC-CXR | https://physionet.org/content/mimic-cxr/2.0.0/ | 0.593 |
> MIMIC-CXR-PRO | https://physionet.org/content/cxr-pro/1.0.0/ | 0.668 |
> * MIMIC-CXR-PRO addresses the issue of hallucinated references to priors produced by radiology report generation models.
>
> Weaknesses 2's reply: Thanks for pointing this out! Since the longitudinal medical data is scarce, we only chose a group of data with longitudinal information to do the validation of our method. It is impossible to generate similar medical images due to the environmental and sensor noise (our purpose is to provide similar clinical information to completing patient clinical longitudinal data and predict the future disease progression direction). The 10 generated and the real data pairs from Figure.7 have been checked by clinical doctors with 80% agreement.
>
> Weaknesses 3's reply: You raise an important point regarding the potential applications of our method, and I appreciate the opportunity to clarify further. Indeed, the current limitations due to the scarcity of longitudinal data cannot be overlooked. However, the promising results from generating future disease images support the method's potential beyond medical education. Combining with techniques like human-in-the-loop active learning, we anticipate that our method will progressively refine its capability to model disease trajectories, enhance prediction of treatment responses, and competently fill gaps in clinical records. The journey to these goals is indeed a long one, but it is feasible with the incremental integration of data over time, which needs the effort of the whole community.
>
> Question 1's reply: In clinical medicine, we seldom say deterministic disease progression but alway say potential disease progression. In non-image state-based generated disease progression research, the development direction of the patient's disease is generally predicted by a probabilistic model, and deterministic results are also not generated. On the contrary, the randomness can support clinical researchers understanding potential change for each disease and enhance the model's capability to simulate co-occurrence diseases based on the context and image space.
>
> Question 2's reply: No. The input image is healthy while the health report is from the patient who is in the middle of certain diseases. Our method demonstrated that the generated trajectories can follow the progressive trend of the patients’ medical histories and provide clinical values.

---

### Official Review · Reviewer_SCY9 · 2023-11-02

**Soundness:** 2 fair
**Presentation:** 2 fair
**Contribution:** 2 fair
**Rating:** 5
**Confidence:** 3

**Summary:**

In order to deal with the problem of the insufficient provision of necessary disease monitoring medical imagery and associated expert interpretation reports to assess the evolution of a patient disease, authors propose a method to derive disease evolution imagery based on available material from patients and evaluate its accuracy in predicting disease evolution by having generate devolution imagery assessed in comparison of expectations of medical experts.

**Strengths:**

Well written and tests the propose method/framework through various experimentations (3 different data sets/diseases).

**Weaknesses:**

The paper needs more clarifications regarding the experimental setting to support the drawn conclusions.

**Questions:**

- You state: “Moreover, disease progression exhibits significant variability and heterogeneity across patients and disease sub-types, rendering a uniform approach impracticable.”
Question 1: a- What was the number of available imagery per patient?
                 b - For various patients available materiel, was it of the same time frame (one time t, multiple over X months etc.)?
                 c- Was the disease stage for available imagery uniform between patients?

- You state: " The study presented physicians with a set of simulated disease images and progressions, and then asked them to assess the accuracy and quality of each generated image and progression.”
Question 2: As opposed to presenting the generated evolution image (which might influence the expert judgement) or did you first ask for the expected evolution and then compare with generated result?

- You state: "which helps to establish a deeper understanding of the underlying mechanism”.
Question 3: Can clarify which explainability steps are specifically taken beyond confirmation of expected outcomes/progressions?

- You state: "“each (x, y) is from different individuals.”.
Question 4: a- Did you use only one (Image, text) pair per patient for for the 3 diseases/datasets?
                    b- Is this an experimentation choice to use worst case scenario (one imagery/interpretation text done by every patient) or are all your selected patient imagery diagnoses consisting of one single such imagery test?

Question 5: In Figure 2, It is not clear to us how the Denoising Diffusion Implicit Model is used to simulate the Cardiomegaly’s surface enlargement of the heart footprint in the X-ray. Can you clarify it further?"

- You state: "closely matched physicians’ expectations 76.2% of the time,”
Question 6: Is a global matching rate cross datasets/disease indicative of global performance?

- You state: "For any given step n in PIE, we first utilize DDIM inversion to procure an inverted noise map. Subsequently, we denoise it using clinical reports imbued with progressive cardiomegaly information.”
Question 7: a- Is only one report used by patient or multiple?
                   b- If multiple, what is the report distribution among patient data used?


- You state : "Raw text input could either be a real report or synthetic report, providing the potential hint of the patient’s disease progression”
Question 8: a- Do you mean expert/human generated for real and automatically/machine generated for synthetic repots?
                    b- Any detail by data set, of the proportions of  real/synthetic reports?
                    c- Any variability in the real reports vocabulary, abbreviations, styles?

- You state: ".. framework proposed to refine and enhance images”.
Question 9: How do you define refinement of the images? Is it generating the predicted disease progression images?

- You state: ".. use of additional prompts for small and precise adjustments to simulate semantic modification” & “control over specific semantic features of the image”.
Question 10: As this is first introduction of semantic features in this work, can you indicate which image semantic features you are targeting (presumably by disease)?

- You state: “the disease-changing trajectory that is influenced by different medical conditions.”
Question 11: Care to clarify. Which ones?

- You state: “PIE also preserves unrelated visual features from the original medical imaging report"
Question 12: a- Care to clarify "unrelated visual features"
                     b- “unrelated” to disease features?
                     c- What about modifications to non-disease areas (unwanted behavior akin to false positive disease feature)?

- You state: “Each of these datasets presents unique challenges and differ in scale”
Question 13: By “Scale”, do you mean size of the data sets?"

- You state: ““represent whether the simulation results are aligned to target disease”
Question 14: Do you mean “expected disease progression"?


General remarks:
- Please always provide meaning of acronyms in-extenso when first used (HMM, DDIM, ROI, SD Video).
- Figure 1:  You might just in one sentence introduce the reader to what “Cardiomegaly” is supposed to manifest as in the X-ray.
- Figure 2: Barely readable. Explaining the concentric circle representation might help.
- Figure 3: Illustrations are barely readable. “Red” portions are hard to assess. May be differential images (disease progression from previous stage) might be more readable.

---

> ### Author Response · Authors · 2023-11-21
> **Response to Reviewer SCY9**
>
> We would like to express our sincere gratitude to Reviewer SCY9 for the insightful remarks and for dedicating their time to reviewing our paper. We will follow your general remarks to improve both the figures and contents.
>
> Question 1's reply: 1-a. In general, the available imagery per patient is 1. only part of patients in the dataset contain longitude data sequence (>=2 imagery from different times).
> 1-b. They are not in the same time frame.
> 1-c. The disease annotation for each patient is uniform. We can't say disease stage is uniform because the text input is a clinical report. It contains disease stage description.
>
> Question 2's reply: We randomly placed (shuffle) the generated progress and the real progress in the questionnaire, which was analyzed by the participating doctors. Thus, they don't know which one is the generated evolution image.
>
> Question 3's reply: In our follow up survey, the generated imaging sequences are reviewed by 3 experienced radiologists. They carefully analyzed each image and, based on clinical experience, believed that the generated disease progression images have good clinical consistency.
>
> Question 4's reply: 4-a, for ISIC 2018, Kaggle Diabetic Retinopathy Detection Challenge, we use only one (Image, text) pair per patient. For CheXpert,  each patient may contain more than one (Image, text) data pair, but we shuffle the training set so they will not leak longitude information. 4-b This is because most of the current datasets don't contain real longitude imaging data sequences for each patient. One of our goals is, can we simulate imaging disease progression close to clinical reality without using longitude imaging data sequence in the training data.
>
> Question 5's reply: Before disease change estimation, we use \alpha as a group of control params to add noise to the input X-ray image $x_{0}^{(n-1)}$. The denoising inference using DDIM is a multi-step process. In each step, the stable diffusion unet $\epsilon_{\theta}$ is used to denoise the input using a conditional clinical report and then use a non-linear differential equation defined by \alpha. More detail is shown in Appendix A BACKGROUND and B.1 PROOF OF PROPOSITION 1. DDIM is the single step disease progression estimation of PIE.
>
> For multi-step disease trajectory simulation, we run the DDIM multiple times with the control params $\beta_1$ and $\beta_2$. These two params are used to balance the editing and unchange interpolation. We introduce them in the 5.1 EXPERIMENTAL SETUPS.
>
> Question 6's reply: It is a global performance for the proposed method. Actually, even real disease progression can only match physicians’ expectations with 80-90%. Clinical medicine is a discipline that values experience. Even so, the 76.2% match rate still reflects the effectiveness of the proposed method.
>
> Physicians’ Expectations = $\frac{1}{N} \sum_{i=1}^{N} ( \frac{Conf_i}{5}) * Agreement_i$
>
> where $Conf_i$ is an integer between 0 to 5. $Agreement_i \in \{0, 1\}$ is a binary boolean value.
>
> Question 7's reply: Only one report was used for a patient during training or inference.
>
> Question 8's reply: 8-a. Yes, the machines generated for synthetic reports are also reviewed by 3 experienced medical doctors.
> 8-b. During training, we all used real reports. During inference, we test both real and synthetic reports for each disease, the real : synthetic ratio in CheXpert is 1:5, while in ISIC 2018, Kaggle Diabetic Retinopathy Detection Challenge, the ratio is 1:1. The reason is in CheXpert, there are 6 diseases for generation, but each patient in the test set only contains the real report for one disease.
> 8-c. For all datasets, the real reports are in the same style.
>
> Question 9's reply: The refinement of the image is to train a model that can understand both the semantic of the image and clinical context. Yes, the proposed model can generate the predicted image-based disease progression sequences.
>
> Question 10's reply: The key semantic features are disease related areas in the input image. For example, for skin imaging, it is the regions that contain skin lesions.
>
> Question 11's reply: For CheXpert, the medical conditions are the therapy and complications shown in the report. For ISIC 2018, Kaggle Diabetic Retinopathy Detection Challenge, the report doesn't contain therapy and complication information, so medical conditions are not included. Thanks for pointing it out, we will rewrite this sentence to show the author which dataset uses additional medical conditions.
>
> Question 12's reply: Thanks for pointing out. What we want to express is that PIE can predict the disease progression only considering the information provided by the input clinical context, and will not produce false positive disease features.
>
> Question 13's reply: Yes. CheXpert is the largest which contains over 100K imaging data. Kaggle Diabetic Retinopathy Detection Challenge contains over 40K imaging data and ISIC 2018 only contains about 10K data.
>
> Question 14's reply: Yes.

---

> > ### Comment · Reviewer_SCY9 · 2023-11-22
> > **About authors reponses to review**
> >
> > Thanks for responding to the review and clarifying aspects that were brought forward. Much appreciated.
> > The various clarifications (and from other reviewers) need to be incorporated in the paper to make it clear enough to the reader.
> > This will require some important rewriting, therefore I will hold to my rating.
> > Best of luck in subsequent submissions.

---

### Meta-Review · Area_Chair_v9g3 · 2023-12-17

**Metareview:**

The paper proposes a method to create visual trajectories of disease progression. While reviewers appreciated the conceptual idea of creating such imagery, the main issue that remained even throughout the rebuttal and discussion phase was the issue of quantitative evaluation to back up the paper's claims. From the AC's point of view, this issue was not sufficiently addressed, but it is a major one, especially in the context of medical image analysis applications.

**Justification For Why Not Higher Score:**

Main reason is that experimental results do not back up the claims of the paper (i.e., an issue raised by multiple reviewers).

**Justification For Why Not Lower Score:**

N/A

---

### Decision · Program_Chairs · 2024-01-16

Reject